# LEVERAGING UNLABELED DATA TO PREDICT OUT-OF-DISTRIBUTION PERFORMANCE

**Saurabh Garg**\*
Carnegie Mellon University
sgarg2@andrew.cmu.edu

**Sivaraman Balakrishnan**
Carnegie Mellon University
sbalakri@andrew.cmu.edu

**Zachary C. Lipton**
Carnegie Mellon University
zlipton@andrew.cmu.edu

**Behnam Neyshabur**
Google Research, Blueshift team
neyshabur@google.com

**Hanie Sedghi**
Google Research, Brain team
hsedghi@google.com

## ABSTRACT

Real-world machine learning deployments are characterized by mismatches between the source (training) and target (test) distributions that may cause performance drops. In this work, we investigate methods for predicting the target domain accuracy using only labeled source data and unlabeled target data. We propose Average Thresholded Confidence (ATC), a practical method that learns a *threshold* on the model's confidence, predicting accuracy as the fraction of unlabeled examples for which model confidence exceeds that threshold. ATC outperforms previous methods across several model architectures, types of distribution shifts (e.g., due to synthetic corruptions, dataset reproduction, or novel subpopulations), and datasets (WILDS, ImageNet, BREEDS, CIFAR, and MNIST). In our experiments, ATC estimates target performance 2–4$\times$ more accurately than prior methods. We also explore the theoretical foundations of the problem, proving that, in general, identifying the accuracy is just as hard as identifying the optimal predictor and thus, the efficacy of any method rests upon (perhaps unstated) assumptions on the nature of the shift. Finally, analyzing our method on some toy distributions, we provide insights concerning when it works.

## 1 INTRODUCTION

Machine learning models deployed in the real world typically encounter examples from previously unseen distributions. While the IID assumption enables us to evaluate models using held-out data from the *source* distribution (from which training data is sampled), this estimate is no longer valid in presence of a distribution shift. Moreover, under such shifts, model accuracy tends to degrade (Szegedy et al., 2014; Recht et al., 2019; Koh et al., 2021). Commonly, the only data available to the practitioner are a labeled training set (source) and unlabeled deployment-time data which makes the problem more difficult. In this setting, detecting shifts in the distribution of covariates is known to be possible (but difficult) in theory (Ramdas et al., 2015), and in practice (Rabanser et al., 2018). However, producing an optimal predictor using only labeled source and unlabeled target data is well-known to be impossible absent further assumptions (Ben-David et al., 2010; Lipton et al., 2018).

Two vital questions that remain are: (i) the precise conditions under which we can estimate a classifier's target-domain accuracy; and (ii) which methods are most practically useful. To begin, the straightforward way to assess the performance of a model under distribution shift would be to collect labeled (target domain) examples and then to evaluate the model on that data. However, collecting fresh labeled data from the target distribution is prohibitively expensive and time-consuming, especially if the target distribution is non-stationary. Hence, instead of using labeled data, we aim to use unlabeled data from the target distribution, that is comparatively abundant, to predict model performance. Note that in this work, our focus is *not* to improve performance on the target but, rather, to estimate the accuracy on the target for a given classifier.

---

\*Work done in part while Saurabh Garg was interning at Google

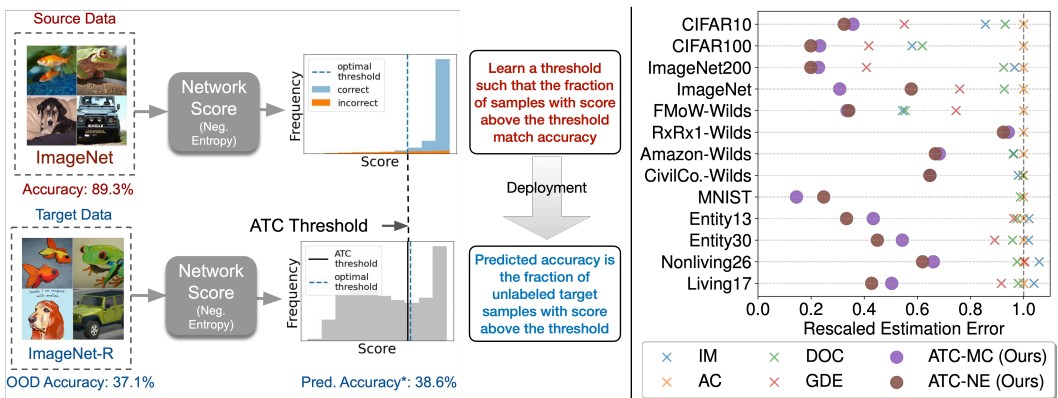

Figure 1: *Illustration of our proposed method ATC.* **Left**: using source domain validation data, we identify a *threshold* on a score (e.g. negative entropy) computed on model confidence such that fraction of examples above the threshold matches the validation set accuracy. ATC estimates accuracy on unlabeled target data as the fraction of examples with the score above the threshold. Interestingly, this threshold yields accurate estimates on a wide set of target distributions resulting from natural and synthetic shifts. **Right**: Efficacy of ATC over previously proposed approaches on our testbed with a post-hoc calibrated model. To obtain errors on the same scale, we rescale all errors with Average Confidence (AC) error. Lower estimation error is better. See Table 1 for exact numbers and comparison on various types of distribution shift. See Sec. 5 for details on our testbed.

Recently, numerous methods have been proposed for this purpose (Deng & Zheng, 2021; Chen et al., 2021; Jiang et al., 2021; Deng et al., 2021; Guillory et al., 2021). These methods either require calibration on the target domain to yield consistent estimates (Jiang et al., 2021; Guillory et al., 2021) or additional labeled data from several target domains to learn a linear regression function on a distributional distance that then predicts model performance (Deng et al., 2021; Deng & Zheng, 2021; Guillory et al., 2021). However, methods that require calibration on the target domain typically yield poor estimates since deep models trained and calibrated on source data are not, in general, calibrated on a (previously unseen) target domain (Ovadia et al., 2019). Besides, methods that leverage labeled data from target domains rely on the fact that unseen target domains exhibit strong linear correlation with seen target domains on the underlying distance measure and, hence, can be rendered ineffective when such target domains with labeled data are unavailable (in Sec. 5.1 we demonstrate such a failure on a real-world distribution shift problem). Therefore, throughout the paper, we assume access to labeled source data and only unlabeled data from target domain(s).

In this work, we first show that absent assumptions on the source classifier or the nature of the shift, no method of estimating accuracy will work generally (even in non-contrived settings). To estimate accuracy on target domain *perfectly*, we highlight that even given perfect knowledge of the labeled source distribution (i.e., $p_s(x, y)$) and unlabeled target distribution (i.e., $p_t(x)$), we need restrictions on the nature of the shift such that we can uniquely identify the target conditional $p_t(y|x)$. Thus, in general, identifying the accuracy of the classifier is as hard as identifying the optimal predictor.

Second, motivated by the superiority of methods that use maximum softmax probability (or logit) of a model for Out-Of-Distribution (OOD) detection (Hendrycks & Gimpel, 2016; Hendrycks et al., 2019), we propose a simple method that leverages softmax probability to predict model performance. Our method, Average Thresholded Confidence (ATC), learns a threshold on a score (e.g., maximum confidence or negative entropy) of model confidence on validation source data and predicts target domain accuracy as the fraction of unlabeled target points that receive a score above that threshold. ATC selects a threshold on validation source data such that the fraction of source examples that receive the score above the threshold match the accuracy of those examples. Our primary contribution in ATC is the proposal of obtaining the threshold and observing its efficacy on (practical) accuracy estimation. Importantly, our work takes a step forward in positively answering the question raised in Deng & Zheng (2021); Deng et al. (2021) about a practical strategy to select a threshold that enables accuracy prediction with thresholded model confidence.

ATC is simple to implement with existing frameworks, compatible with arbitrary model classes, and dominates other contemporary methods. Across several model architectures on a range of benchmark

vision and language datasets, we verify that ATC outperforms prior methods by at least 2–4× in predicting target accuracy on a variety of distribution shifts. In particular, we consider shifts due to common corruptions (e.g., ImageNet-C), natural distribution shifts due to dataset reproduction (e.g., ImageNet-v2, ImageNet-R), shifts due to novel subpopulations (e.g., BREEDS), and distribution shifts faced in the wild (e.g., WILDS).

As a starting point for theory development, we investigate ATC on a simple toy model that models distribution shift with varying proportions of the population with spurious features, as in Nagarajan et al. (2020). Finally, we note that although ATC achieves superior performance in our empirical evaluation, like all methods, it must fail (returns inconsistent estimates) on certain types of distribution shifts, per our impossibility result.

## 2 PRIOR WORK

**Out-of-distribution detection.** The main goal of OOD detection is to identify previously unseen examples, i.e., samples out of the support of training distribution. To accomplish this, modern methods utilize confidence or features learned by a deep network trained on some source data. Hendrycks & Gimpel (2016); Geifman & El-Yaniv (2017) used the confidence score of an (already) trained deep model to identify OOD points. Lakshminarayanan et al. (2016) use entropy of an ensemble model to evaluate prediction uncertainty on OOD points. To improve OOD detection with model confidence, Liang et al. (2017) propose to use temperature scaling and input perturbations. Jiang et al. (2018) propose to use scores based on the relative distance of the predicted class to the second class. Recently, residual flow-based methods were used to obtain a density model for OOD detection (Zhang et al., 2020). Ji et al. (2021) proposed a method based on subfunction error bounds to compute unreliability per sample. Refer to Ovadia et al. (2019); Ji et al. (2021) for an overview and comparison of methods for prediction uncertainty on OOD data.

**Predicting model generalization.** Understanding generalization capabilities of overparameterized models on in-distribution data using conventional machine learning tools has been a focus of a long line of work; representative research includes Neyshabur et al. (2015; 2017); Neyshabur (2017); Neyshabur et al. (2018); Dziugaite & Roy (2017); Bartlett et al. (2017); Zhou et al. (2018); Long & Sedghi (2019); Nagarajan & Kolter (2019a). At a high level, this line of research bounds the generalization gap directly with complexity measures calculated on the trained model. However, these bounds typically remain numerically loose relative to the true generalization error (Zhang et al., 2016; Nagarajan & Kolter, 2019b). On the other hand, another line of research departs from complexity-based approaches to use unseen unlabeled data to predict in-distribution generalization (Platanios et al., 2016; 2017; Garg et al., 2021; Jiang et al., 2021).

Relevant to our work are methods for predicting the error of a classifier on OOD data based on unlabeled data from the target (OOD) domain. These methods can be characterized into two broad categories: (i) Methods which explicitly predict correctness of the model on individual unlabeled points (Deng & Zheng, 2021; Jiang et al., 2021; Deng et al., 2021); and (ii) Methods which directly obtain an estimate of error with unlabeled OOD data without making a point-wise prediction (Chen et al., 2021; Guillory et al., 2021; Chuang et al., 2020).

To achieve a consistent estimate of the target accuracy, Jiang et al. (2021); Guillory et al. (2021) require calibration on target domain. However, these methods typically yield poor estimates as deep models trained and calibrated on some source data are seldom calibrated on previously unseen domains (Ovadia et al., 2019). Additionally, Deng & Zheng (2021); Guillory et al. (2021) derive model-based distribution statistics on unlabeled target set that correlate with the target accuracy and propose to use a subset of *labeled* target domains to learn a (linear) regression function that predicts model performance. However, there are two drawbacks with this approach: (i) the correlation of these distribution statistics can vary substantially as we consider different nature of shifts (refer to Sec. 5.1, where we empirically demonstrate this failure); (ii) even if there exists a (hypothetical) statistic with strong correlations, obtaining labeled target domains (even simulated ones) with strong correlations would require significant *a priori* knowledge about the nature of shift that, in general, might not be available before models are deployed in the wild. Nonetheless, in our work, we only assume access to labeled data from the source domain presuming no access to labeled target domains or information about how to simulate them.

Moreover, unlike the parallel work of Deng et al. (2021), we do not focus on methods that alter the training on source data to aid accuracy prediction on the target data. Chen et al. (2021) propose an

importance re-weighting based approach that leverages (additional) information about the axis along which distribution is shifting in form of "slicing functions". In our work, we make comparisons with importance re-weighting baseline from Chen et al. (2021) as we do not have any additional information about the axis along which the distribution is shifting.

## 3 PROBLEM SETUP

**Notation.** By $\|\cdot\|$, and $\langle\cdot,\cdot\rangle$ we denote the Euclidean norm and inner product, respectively. For a vector $v \in \mathbb{R}^d$, we use $v_j$ to denote its $j^{\text{th}}$ entry, and for an event $E$ we let $\mathbb{I}\left[E\right]$ denote the binary indicator of the event.

Suppose we have a multi-class classification problem with the input domain $\mathcal{X} \subseteq \mathbb{R}^d$ and label space $\mathcal{Y} = \{1, 2, \ldots, k\}$. For binary classification, we use $\mathcal{Y} = \{0, 1\}$. By $\mathcal{D}^{\text{S}}$ and $\mathcal{D}^{\text{T}}$, we denote source and target distribution over $\mathcal{X} \times \mathcal{Y}$. For distributions $\mathcal{D}^{\text{S}}$ and $\mathcal{D}^{\text{T}}$, we define $p_{\text{S}}$ or $p_{\text{T}}$ as the corresponding probability density (or mass) functions. A dataset $S := \{(x_i, y_i)\}_{i=1}^n \sim (\mathcal{D}^{\text{S}})^n$ contains $n$ points sampled i.i.d. from $\mathcal{D}^{\text{S}}$. Let $\mathcal{F}$ be a class of hypotheses mapping $\mathcal{X}$ to $\Delta^{k-1}$ where $\Delta^{k-1}$ is a simplex in $k$ dimensions. Given a classifier $f \in \mathcal{F}$ and datum $(x, y)$, we denote the 0-1 error (i.e., classification error) on that point by $\mathcal{E}(f(x), y) := \mathbb{I}\left[y \notin \arg\max_{j \in \mathcal{Y}} f_j(x)\right]$. Given a model $f \in \mathcal{F}$, our goal in this work is to understand the performance of $f$ on $\mathcal{D}^{\text{T}}$ without access to labeled data from $\mathcal{D}^{\text{T}}$. Note that our goal is not to adapt the model to the target data. Concretely, we aim to predict accuracy of $f$ on $\mathcal{D}^{\text{T}}$. Throughout this paper, we assume we have access to the following: (i) model $f$; (ii) previously-unseen (validation) data from $\mathcal{D}^{\text{S}}$; and (iii) unlabeled data from target distribution $\mathcal{D}^{\text{T}}$.

### 3.1 ACCURACY ESTIMATION: POSSIBILITY AND IMPOSSIBILITY RESULTS

First, we investigate the question of when it is possible to estimate the target accuracy of an arbitrary classifier, even given knowledge of the full source distribution $p_s(x, y)$ and target marginal $p_t(x)$. Absent assumptions on the nature of shift, estimating target accuracy is impossible. Even given access to $p_s(x, y)$ and $p_t(x)$, the problem is fundamentally unidentifiable because $p_t(y|x)$ can shift arbitrarily. In the following proposition, we show that absent assumptions on the classifier $f$ (i.e., when $f$ can be any classifier in the space of all classifiers on $\mathcal{X}$), we can estimate accuracy on the target data iff assumptions on the nature of the shift, together with $p_s(x, y)$ and $p_t(x)$, uniquely identify the (unknown) target conditional $p_t(y|x)$. We relegate proofs from this section to App. A.

**Proposition 1.** *Absent further assumptions, accuracy on the target is identifiable iff $p_t(y|x)$ is uniquely identified given $p_s(x, y)$ and $p_t(x)$.*

Proposition 1 states that we need enough constraints on nature of shift such that $p_s(x, y)$ and $p_t(x)$ identifies unique $p_t(y|x)$. It also states that under some assumptions on the nature of the shift, we can hope to estimate the model's accuracy on target data. We will illustrate this on two common assumptions made in domain adaptation literature: (i) covariate shift (Heckman, 1977; Shimodaira, 2000) and (ii) label shift (Saerens et al., 2002; Zhang et al., 2013; Lipton et al., 2018). Under covariate shift assumption, that the target marginal support $\mathbf{supp}(p_t(x))$ is a subset of the source marginal support $\mathbf{supp}(p_s(x))$ and that the conditional distribution of labels given inputs does not change within support, i.e., $p_s(y|x) = p_t(y|x)$, which, trivially, identifies a unique target conditional $p_t(y|x)$. Under label shift, the reverse holds, i.e., the class-conditional distribution does not change ($p_s(x|y) = p_t(x|y)$) and, again, information about $p_t(x)$ uniquely determines the target conditional $p_t(y|x)$ (Lipton et al., 2018; Garg et al., 2020). In these settings, one can estimate an arbitrary classifier's accuracy on the target domain either by using importance re-weighting with the ratio $p_t(x)/p_s(x)$ in case of covariate shift or by using importance re-weighting with the ratio $p_t(y)/p_s(y)$ in case of label shift. While importance ratios in the former case can be obtained directly when $p_t(x)$ and $p_s(x)$ are known, the importance ratios in the latter case can be obtained by using techniques from Lipton et al. (2018); Azizzadenesheli et al. (2019); Alexandari et al. (2019).

As a corollary of Proposition 1, we now present a simple impossibility result, demonstrating that no single method can work for all families of distribution shift.

**Corollary 1.** *Absent assumptions on the classifier $f$, no method of estimating accuracy will work in all scenarios, i.e., for different nature of distribution shifts.*

Intuitively, this result states that every method of estimating accuracy on target data is tied up with some assumption on the nature of the shift and might not be useful for estimating accuracy under

a different assumption on the nature of the shift. For illustration, consider a setting where we have access to distribution $p_s(x, y)$ and $p_t(x)$. Additionally, assume that the distribution can shift only due to covariate shift or label shift without any knowledge about which one. Then Corollary 1 says that it is impossible to have a single method that will simultaneously for both label shift and covariate shift as in the following example (we spell out the details in App. A):

**Example 1.** Assume binary classification with $p_s(x) = \alpha \cdot \phi(\mu_1) + (1 - \alpha) \cdot \phi(\mu_2)$, $p_s(x|y = 0) = \phi(\mu_1)$, $p_s(x|y = 1) = \phi(\mu_2)$, and $p_t(x) = \beta \cdot \phi(\mu_1) + (1 - \beta) \cdot \phi(\mu_2)$ where $\phi(\mu) = \mathcal{N}(\mu, 1)$, $\alpha, \beta \in (0, 1)$, and $\alpha \neq \beta$. Error of a classifier $f$ on target data is given by $\mathcal{E}_1 = \mathbb{E}_{(x,y) \sim p_s(x,y)} \left[ \frac{p_t(x)}{p_s(x)} \mathbb{I} \left[ f(x) \neq y \right] \right]$ under covariate shift and by $\mathcal{E}_2 = \mathbb{E}_{(x,y) \sim p_s(x,y)} \left[ \left( \frac{\beta}{\alpha} \mathbb{I} \left[ y = 0 \right] + \frac{1-\beta}{1-\alpha} \mathbb{I} \left[ y = 1 \right] \right) \mathbb{I} \left[ f(x) \neq y \right] \right]$ under label shift. In App. A, we show that $\mathcal{E}_1 \neq \mathcal{E}_2$ for all $f$. Thus, given access to $p_s(x, y)$, and $p_t(x)$, any method that consistently estimates error of a classifier under covariate shift will give an incorrect estimate of error under label shift and vice-versa. The reason is that the same $p_t(x)$ and $p_s(x, y)$ can correspond to error $\mathcal{E}_1$ (under covariate shift) or error $\mathcal{E}_2$ (under label shift) and determining which scenario one faces requires further assumptions on the nature of shift.

## 4 Predicting accuracy with Average Thresholded Confidence

In this section, we present our method ATC that leverages a black box classifier $f$ and (labeled) validation source data to predict accuracy on target domain given access to unlabeled target data. Throughout the discussion, we assume that the classifier $f$ is fixed.

Before presenting our method, we introduce some terminology. Define a score function $s : \Delta^{k-1} \to \mathbb{R}$ that takes in the softmax prediction of the function $f$ and outputs a scalar. We want a score function such that if the score function takes a high value at a datum $(x, y)$ then $f$ is likely to be correct. In this work, we explore two such score functions: (i) Maximum confidence, i.e., $s(f(x)) = \max_{j \in \mathcal{Y}} f_j(x)$; and (ii) Negative Entropy, i.e., $s(f(x)) = \sum_j f_j(x) \log(f_j(x))$. Our method identifies a threshold $t$ on source data $\mathcal{D}^S$ such that the expected number of points that obtain a score less than $t$ match the error of $f$ on $\mathcal{D}^S$, i.e.,

$$\mathbb{E}_{x \sim \mathcal{D}^S} \left[ \mathbb{I} \left[ s(f(x)) < t \right] \right] = \mathbb{E}_{(x,y) \sim \mathcal{D}^S} \left[ \mathbb{I} \left[ \arg\max_{j \in \mathcal{Y}} f_j(x) \neq y \right] \right], \tag{1}$$

and then our error estimate $\text{ATC}_{\mathcal{D}^T}(s)$ on the target domain $\mathcal{D}^T$ is given by the expected number of target points that obtain a score less than $t$, i.e.,

$$\text{ATC}_{\mathcal{D}^T}(s) = \mathbb{E}_{x \sim \mathcal{D}^T} \left[ \mathbb{I} \left[ s(f(x)) < t \right] \right]. \tag{2}$$

In short, in (1), ATC selects a threshold on the score function such that the error in the source domain matches the expected number of points that receive a score below $t$ and in (2), ATC predicts error on the target domain as the fraction of unlabeled points that obtain a score below that threshold $t$. Note that, in principle, there exists a different threshold $t'$ on the target distribution $\mathcal{D}^T$ such that (1) is satisfied on $\mathcal{D}^T$. However, in our experiments, the same threshold performs remarkably well. The main empirical contribution of our work is to show that the threshold obtained with (1) might be used effectively in conduction with modern deep networks in a wide range of settings to estimate error on the target data. In practice, to obtain the threshold with ATC, we minimize the difference between the expression on two sides of (1) using finite samples. In the next section, we show that ATC precisely predicts accuracy on the OOD data on the desired line $y = x$. In App. C, we discuss an alternate interpretation of the method and make connections with OOD detection methods.

## 5 Experiments

We now empirical evaluate ATC and compare it with existing methods. In each of our main experiment, keeping the underlying model fixed, we vary target datasets and make a prediction of the target accuracy with various methods given access to only unlabeled data from the target. Unless noted otherwise, all models are trained only on samples from the source distribution with the main exception of pre-training on a different distribution. We use labeled examples from the target distribution to only obtain true error estimates.

**Datasets.** First, we consider synthetic shifts induced due to different visual corruptions (e.g., shot noise, motion blur etc.) under ImageNet-C (Hendrycks & Dietterich, 2019). Next, we consider

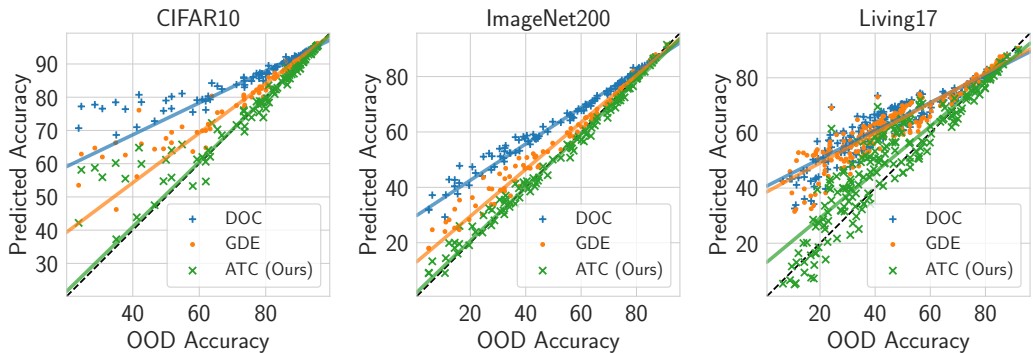

Figure 2: *Scatter plot of predicted accuracy versus (true) OOD accuracy.* Each point denotes a different OOD dataset, all evaluated with the same DenseNet121 model. We only plot the best three methods. With ATC (ours), we refer to ATC-NE. We observe that ATC significantly outperforms other methods and with ATC, we recover the desired line $y = x$ with a robust linear fit. Aggregated estimation error in Table 1 and plots for other datasets and architectures in App. H.

natural shifts due to differences in the data collection process of ImageNet (Russakovsky et al., 2015), e.g, ImageNetv2 (Recht et al., 2019). We also consider images with artistic renditions of object classes, i.e., ImageNet-R (Hendrycks et al., 2021) and ImageNet-Sketch (Wang et al., 2019). Note that renditions dataset only contains a subset 200 classes from ImageNet. To include renditions dataset in our testbed, we include results on ImageNet restricted to these 200 classes (which we call ImageNet-200) along with full ImageNet.

Second, we consider BREEDS (Santurkar et al., 2020) to assess robustness to subpopulation shifts, in particular, to understand how accuracy estimation methods behave when novel subpopulations not observed during training are introduced. BREEDS leverages class hierarchy in ImageNet to create 4 datasets ENTITY-13, ENTITY-30, LIVING-17, NON-LIVING-26. We focus on natural and synthetic shifts as in ImageNet on same and different subpopulations in BREEDs. Third, from WILDS (Koh et al., 2021) benchmark, we consider FMoW-WILDS (Christie et al., 2018), RxRx1-WILDS (Taylor et al., 2019), Amazon-WILDS (Ni et al., 2019), CivilComments-WILDS (Borkan et al., 2019) to consider distribution shifts faced in the wild.

Finally, similar to ImageNet, we consider (i) synthetic shifts (CIFAR-10-C) due to common corruptions; and (ii) natural shift (i.e., CIFARv2 (Recht et al., 2018)) on CIFAR-10 (Krizhevsky & Hinton, 2009). On CIFAR-100, we just have synthetic shifts due to common corruptions. For completeness, we also consider natural shifts on MNIST (LeCun et al., 1998) as in the prior work (Deng & Zheng, 2021). We use three real shifted datasets, i.e., USPS (Hull, 1994), SVHN (Netzer et al., 2011) and QMNIST (Yadav & Bottou, 2019). We give a detailed overview of our setup in App. F.

**Architectures and Evaluation.** For ImageNet, BREEDS, CIFAR, FMoW-WILDS, RxRx1-WILDS datasets, we use DenseNet121 (Huang et al., 2017) and ResNet50 (He et al., 2016) architectures. For Amazon-WILDS and CivilComments-WILDS, we fine-tune a DistilBERT-base-uncased (Sanh et al., 2019) model. For MNIST, we train a fully connected multilayer perceptron. We use standard training with benchmarked hyperparameters. To compare methods, we report average absolute difference between the true accuracy on the target data and the estimated accuracy on the same unlabeled examples. We refer to this metric as Mean Absolute estimation Error (MAE). Along with MAE, we also show scatter plots to visualize performance at individual target sets. Refer to App. G for additional details on the setup.

**Methods** With ATC-NE, we denote ATC with negative entropy score function and with ATC-MC, we denote ATC with maximum confidence score function. For all methods, we implement *post-hoc* calibration on validation source data with Temperature Scaling (TS; Guo et al. (2017)). Below we briefly discuss baselines methods compared in our work and relegate details to App. E.

*Average Confidence (AC).* Error is estimated as the expected value of the maximum softmax confidence on the target data, i.e, $\text{AC}_{\mathcal{D}^\text{T}} = \mathbb{E}_{x \sim \mathcal{D}^\text{T}} \left[ \max_{j \in \mathcal{Y}} f_j(x) \right]$.

*Difference Of Confidence (DOC).* We estimate error on target by subtracting difference of confidences on source and target (as a surrogate to distributional distance Guillory et al. (2021)) from the error on source distribution, i.e, $\text{DOC}_{\mathcal{D}^\text{T}} = \mathbb{E}_{x \sim \mathcal{D}^\text{S}} \left[ \mathbb{I} \left[ \arg\max_{j \in \mathcal{Y}} f_j(x) \neq y \right] \right] + \mathbb{E}_{x \sim \mathcal{D}^\text{T}} \left[ \max_{j \in \mathcal{Y}} f_j(x) \right] - \mathbb{E}_{x \sim \mathcal{D}^\text{S}} \left[ \max_{j \in \mathcal{Y}} f_j(x) \right]$. This is referred to as DOC-Feat in (Guillory et al., 2021).

| Dataset | Shift | IM | | AC | | DOC | | GDE | ATC-MC (Ours) | | ATC-NE (Ours) | |
|---|---|---|---|---|---|---|---|---|---|---|---|---|
| | | Pre T | Post T | Pre T | Post T | Pre T | Post T | Post T | Pre T | Post T | Pre T | Post T |
| CIFAR10 | Natural | 6.60 | 5.74 | 9.88 | 6.89 | 7.25 | 6.07 | 4.77 | 3.21 | 3.02 | 2.99 | **2.85** |
| | Synthetic | 12.33 | 10.20 | 16.50 | 11.91 | 13.87 | 11.08 | 6.55 | 4.65 | 4.25 | 4.21 | **3.87** |
| CIFAR100 | Synthetic | 13.69 | 11.51 | 23.61 | 13.10 | 14.60 | 10.14 | 9.85 | 5.50 | **4.75** | 4.72 | 4.94 |
| ImageNet200 | Natural | 12.37 | 8.19 | 22.07 | 8.61 | 15.17 | 7.81 | 5.13 | 4.37 | 2.04 | 3.79 | **1.45** |
| | Synthetic | 19.86 | 12.94 | 32.44 | 13.35 | 25.02 | 12.38 | 5.41 | 5.93 | 3.09 | 5.00 | **2.68** |
| ImageNet | Natural | 7.77 | 6.50 | 18.13 | 6.02 | 8.13 | 5.76 | 6.23 | 3.88 | 2.17 | 2.06 | **0.80** |
| | Synthetic | 13.39 | 10.12 | 24.62 | 8.51 | 13.55 | 7.90 | 6.32 | 3.34 | **2.53** | 2.61 | 4.89 |
| FMoW-WILDS | Natural | 5.53 | 4.31 | 33.53 | 12.84 | 5.94 | 4.45 | 5.74 | 3.06 | **2.70** | 3.02 | 2.72 |
| RxRx1-WILDS | Natural | 5.80 | 5.72 | 7.90 | 4.84 | 5.98 | 5.98 | 6.03 | 4.66 | **4.56** | 4.41 | 4.47 |
| Amazon-WILDS | Natural | 2.40 | 2.29 | 8.01 | 2.38 | 2.40 | 2.28 | 17.87 | 1.65 | **1.62** | 1.60 | 1.59 |
| CivilCom.-WILDS | Natural | 12.64 | 10.80 | 16.76 | 11.03 | 13.31 | 10.99 | 16.65 | | **7.14** | | |
| MNIST | Natural | 18.48 | 15.99 | 21.17 | 14.81 | 20.19 | 14.56 | 24.42 | 5.02 | **2.40** | 3.14 | 3.50 |
| ENTITY-13 | Same | 16.23 | 11.14 | 24.97 | 10.88 | 19.08 | 10.47 | 10.71 | 5.39 | **3.88** | 4.58 | 4.19 |
| | Novel | 28.53 | 22.02 | 38.33 | 21.64 | 32.43 | 21.22 | 20.61 | 13.58 | 10.28 | 12.25 | **6.63** |
| ENTITY-30 | Same | 18.59 | 14.46 | 28.82 | 14.30 | 21.63 | 13.46 | 12.92 | 9.12 | **7.75** | 8.15 | 7.64 |
| | Novel | 32.34 | 26.85 | 44.02 | 26.27 | 36.82 | 25.42 | 23.16 | 17.75 | 14.30 | 15.60 | **10.57** |
| NONLIVING-26 | Same | 18.66 | 17.17 | 26.39 | 16.14 | 19.86 | 15.58 | 16.63 | 10.87 | **10.24** | 10.07 | 10.26 |
| | Novel | 33.43 | 31.53 | 41.66 | 29.87 | 35.13 | 29.31 | 29.56 | 21.70 | 20.12 | 19.08 | **18.26** |
| LIVING-17 | Same | 12.63 | 11.05 | 18.32 | 10.46 | 14.43 | 10.14 | 9.87 | 4.57 | **3.95** | 3.81 | 4.21 |
| | Novel | 29.03 | 26.96 | 35.67 | 26.11 | 31.73 | 25.73 | 23.53 | 16.15 | 14.49 | 12.97 | **11.39** |

Table 1: *Mean Absolute estimation Error (MAE) results for different datasets in our setup grouped by the nature of shift.* 'Same' refers to same subpopulation shifts and 'Novel' refers novel subpopulation shifts. We include details about the target sets considered in each shift in Table 2. Post T denotes use of TS calibration on source. Across all datasets, we observe that ATC achieves superior performance (lower MAE is better). For language datasets, we use DistilBERT-base-uncased, for vision dataset we report results with DenseNet model with the exception of MNIST where we use FCN. We include results on other architectures in App. H. For GDE post T and pre T estimates match since TS doesn't alter the argmax prediction. Results reported by aggregating MAE numbers over 4 different seeds. We include results with standard deviation values in Table 3.

*Importance re-weighting (IM).* We estimate the error of the classifier with importance re-weighting of 0-1 error in the pushforward space of the classifier. This corresponds to MANDOLIN using one slice based on the underlying classifier confidence Chen et al. (2021).

*Generalized Disagreement Equality (GDE).* Error is estimated as the expected disagreement of two models (trained on the same training set but with different randomization) on target data (Jiang et al., 2021), i.e., $\text{GDE}_{\mathcal{D}^\text{T}} = \mathbb{E}_{x \sim \mathcal{D}^\text{T}} \left[ \mathbb{I} \left[ f(x) \neq f'(x) \right] \right]$ where $f$ and $f'$ are the two models. Note that GDE requires two models trained independently, doubling the computational overhead while training.

## 5.1 RESULTS

In Table 1, we report MAE results aggregated by the nature of the shift in our testbed. In Fig. 2 and Fig. 1(right), we show scatter plots for predicted accuracy versus OOD accuracy on several datasets. We include scatter plots for all datasets and parallel results with other architectures in App. H. In App. H.1, we also perform ablations on CIFAR using a pre-trained model and observe that pre-training doesn't change the efficacy of ATC.

We predict accuracy on the target data before and after calibration with TS. First, we observe that both ATC-NE and ATC-MC (even without TS) obtain significantly lower MAE when compared with other methods (even with TS). Note that with TS we observe substantial improvements in MAE for all methods. Overall, ATC-NE (with TS) typically achieves the smallest MAE improving by more than $2\times$ on CIFAR and by $3$–$4\times$ on ImageNet over GDE (the next best alternative to ATC). Alongside, we also observe that a linear fit with robust regression (Siegel, 1982) on the scatter plot recovers a line close to $x = y$ for ATC-NE with TS while the line is far away from $x = y$ for other

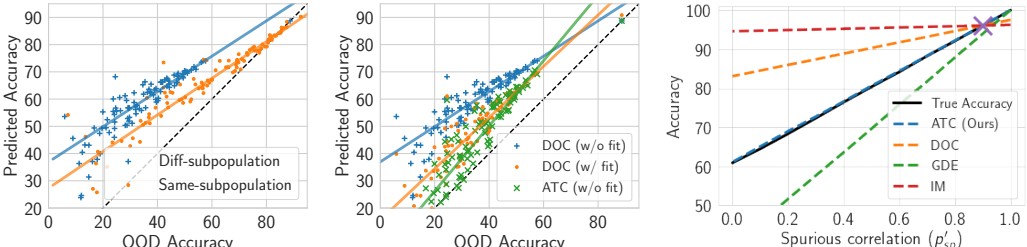

Figure 3: **Left:** Predicted accuracy with DOC on Living17 BREEDS dataset. We observe a substantial gap in the linear fit of same and different subpopulations highlighting poor correlation. **Middle:** After fitting a robust linear model for DOC on same subpopulation, we show predicted accuracy on different subpopulations with fine-tuned DOC (i.e., DOC (w/ fit)) and compare with ATC without any regression model, i.e., ATC (w/o fit). While observe substantial improvements in MAE from 24.41 with DOC (w/o fit) to 13.26 with DOC (w/ fit), ATC (w/o fit) continues to outperform even DOC (w/ fit) with MAE 10.22. We show parallel results with other BREEDS datasets in App. H.2. **Right :** Empirical validation of our toy model. We show that ATC perfectly estimates target performance as we vary the degree of spurious correlation in target. '×' represents accuracy on source.

methods (Fig. 2 and Fig. 1(right)). Remarkably, MAE is in the range of 0.4–5.8 with ATC for CIFAR, ImageNet, MNIST, and Wilds. However, MAE is much higher on BREEDS benchmark with novel subpopulations. While we observe a small MAE (i.e., comparable to our observations on other datasets) on BREEDS with natural and synthetic shifts from the same sub-population, MAE on shifts with novel population is significantly higher with all methods. Note that even on novel populations, ATC continues to dominate all other methods across all datasets in BREEDS.

Additionally, for different subpopulations in BREEDS setup, we observe a poor linear correlation of the estimated performance with the actual performance as shown in Fig. 3 (left)(we notice a similar gap in the linear fit for all other methods). Hence in such a setting, we would expect methods that fine-tune a regression model on labeled target examples from shifts with one subpopulation will perform poorly on shifts with different subpopulations. Corroborating this intuition, next, we show that even after fitting a regression model for DOC on natural and synthetic shifts with source subpopulations, ATC without regression model continues to outperform DOC with regression model on shifts with novel subpopulation.

**Fitting a regression model on BREEDS with DOC.** Using label target data from natural and synthetic shifts for the same subpopulation (same as source), we fit a robust linear regression model (Siegel, 1982) to fine-tune DOC as in Guillory et al. (2021). We then evaluate the fine-tuned DOC (i.e., DOC with linear model) on natural and synthetic shifts from novel subpopulations on BREEDS benchmark. Although we observe significant improvements in the performance of fine-tuned DOC when compared with DOC (without any fine-tuning), ATC without any regression model continues to perform better (or similar) to that of fine-tuned DOC on novel subpopulations (Fig. 3 (middle)). Refer to App. H.2 for details and Table 5 for MAE on BREEDS with regression model.

## 6 INVESTIGATING ATC ON TOY MODEL

In this section, we propose and analyze a simple theoretical model that distills empirical phenomena from the previous section and highlights efficacy of ATC. Here, our aim is not to obtain a general model that captures complicated real distributions on high dimensional input space as the images in ImageNet. Instead to further our understanding, we focus on an *easy-to-learn* binary classification task from Nagarajan et al. (2020) with linear classifiers, that is rich enough to exhibit some of the same phenomena as with deep networks on real data distributions.

Consider a easy-to-learn binary classification problem with two features $x = [x_{\text{inv}}, x_{\text{sp}}] \in \mathbb{R}^2$ where $x_{\text{inv}}$ is fully predictive invariant feature with a margin $\gamma > 0$ and $x_{\text{sp}} \in \{-1, 1\}$ is a spurious feature (i.e., a feature that is correlated but not predictive of the true label). Conditional on $y$, the distribution over $x_{\text{inv}}$ is given as follows: $x_{\text{inv}}|(y = 1) \sim U[\gamma, c]$ and $x_{\text{inv}}|(y = 0) \sim U[-c, -\gamma]$, where $c$ is a fixed constant greater than $\gamma$. For simplicity, we assume that label distribution on source is uniform on $\{-1, 1\}$. $x_{\text{sp}}$ is distributed such that $P_s[x_{\text{sp}} \cdot (2y - 1) > 0] = p_{\text{sp}}$, where $p_{\text{sp}} \in (0.5, 1.0)$ controls the degree of spurious correlation. To model distribution shift, we simulate target data with different degree of spurious correlation, i.e., in target distribution $P_t[x_{\text{sp}} \cdot (2y - 1) > 0] = p'_{\text{sp}} \in [0, 1]$. Note

that here we do not consider shifts in the label distribution but our result extends to arbitrary shifts in the label distribution as well.

In this setup, we examine linear sigmoid classifiers of the form $f(x) = \left[ \frac{1}{1+e^{w^T x}}, \frac{e^{w^T x}}{1+e^{w^T x}} \right]$ where $w = [w_{\text{inv}}, w_{\text{sp}}] \in \mathbb{R}^2$. While there exists a linear classifier with $w = [1, 0]$ that correctly classifies all the points with a margin $\gamma$, Nagarajan et al. (2020) demonstrated that a linear classifier will typically have a dependency on the spurious feature, i.e., $w_{\text{sp}} \neq 0$. They show that due to geometric skews, despite having positive dependencies on the invariant feature, a max-margin classifier trained on finite samples relies on the spurious feature. Refer to App. D for more details on these skews. In our work, we show that given a linear classifier that relies on the spurious feature and achieves a non-trivial performance on the source (i.e., $w_{\text{inv}} > 0$), ATC with maximum confidence score function *consistently* estimates the accuracy on the target distribution.

**Theorem 1** (Informal). *Given any classifier with $w_{inv} > 0$ in the above setting, the threshold obtained in* (1) *together with ATC as in* (2) *with maximum confidence score function obtains a consistent estimate of the target accuracy.*

Consider a classifier that depends positively on the spurious feature (i.e., $w_{\text{sp}} > 0$). Then as the spurious correlation decreases in the target data, the classifier accuracy on the target will drop and vice-versa if the spurious correlation increases on the target data. Theorem 1 shows that the threshold identified with ATC as in (1) remains invariant as the distribution shifts and hence ATC as in (2) will correctly estimate the accuracy with shifting distributions. Next, we illustrate Theorem 1 by simulating the setup empirically. First we pick a arbitrary classifier (which can also be obtained by training on source samples), tune the threshold on hold-out source examples and predict accuracy with different methods as we shift the distribution by varying the degree of spurious correlation.

**Empirical validation and comparison with other methods.** Fig. 3(right) shows that as the degree of spurious correlation varies, our method accurately estimates the target performance where all other methods fail to accurately estimate the target performance. Understandably, due to poor calibration of the sigmoid linear classifier AC, DOC and GDE fail. While in principle IM can perfectly estimate the accuracy on target in this case, we observe that it is highly sensitive to the number bins and choice of histogram binning (i.e., uniform mass or equal width binning). We elaborate more on this in App. D.

**Biased estimation with ATC.** Now we discuss changes in the above setup where ATC yields inconsistent estimates. We assumed that both in source and target $x_{\text{inv}}|y = 1$ is uniform between $[\gamma, c]$ and $x|y = -1$ is uniform between $[-c, -\gamma]$. Shifting the support of target class conditional $p_t(x_{\text{inv}}|y)$ may introduce a bias in ATC estimates, e.g., shrinking the support to $c_1 (< c)$ (while maintaining uniform distribution) in the target will lead to an over-estimation of the target performance with ATC. In App. D.1, we elaborate on this failure and present a general (but less interpretable) classifier dependent distribution shift condition where ATC is guaranteed to yield consistent estimates.

## 7 CONCLUSION AND FUTURE WORK

In this work, we proposed ATC, a simple method for estimating target domain accuracy based on unlabeled target (and labeled source data). ATC achieves remarkably low estimation error on several synthetic and natural shift benchmarks in our experiments. Notably, our work draws inspiration from recent state-of-the-art methods that use softmax confidences below a certain threshold for OOD detection (Hendrycks & Gimpel, 2016; Hendrycks et al., 2019) and takes a step forward in answering questions raised in Deng & Zheng (2021) about the practicality of threshold based methods.

Our distribution shift toy model justifies ATC on an easy-to-learn binary classification task. In our experiments, we also observe that calibration significantly improves estimation with ATC. Since in binary classification, post hoc calibration with TS does not change the effective threshold, in future work, we hope to extend our theoretical model to multi-class classification to understand the efficacy of calibration. Our theory establishes that a classifier's accuracy is not, in general identified, from labeled source and unlabeled target data alone, absent considerable additional constraints on the target conditional $p_t(y|x)$. In light of this finding, we also hope to extend our understanding beyond the simple theoretical toy model to characterize broader sets of conditions under which ATC might be guaranteed to obtain consistent estimates. Finally, we should note that while ATC outperforms previous approaches, it still suffers from large estimation error on datasets with novel populations, e.g., BREEDS. We hope that our findings can lay the groundwork for future work for improving accuracy estimation on such datasets.

**Reproducibility Statement**   We have been careful to ensure that our results are reproducible. We have stored all models and logged all hyperparameters and seeds to facilitate reproducibility. Note that throughout our work, we do not perform any hyperparameter tuning, instead, using benchmarked hyperparameters and training procedures to make our results easy to reproduce. While, we have not released code yet, the appendix provides all the necessary details to replicate our experiments and results. Moreover, we plan to release the code with a revised version of the manuscript.

## ACKNOWLEDGEMENT

Authors would like to thank Ariel Kleiner and Sammy Jerome as the problem formulation and motivation of this paper was highly influenced by initial discussions with them.

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

APPENDIX

# A   PROOFS FROM SEC. 3

Before proving results from Sec. 3, we introduce some notations. Define $\mathcal{E}(f(x), y) :=$ $\mathbb{I}\left[y \notin \arg\max_{j \in \mathcal{Y}} f_j(x)\right]$. We express the *population error* on distribution $\mathcal{D}$ as $\mathcal{E}_\mathcal{D}(f) :=$ $\mathbb{E}_{(x,y) \sim \mathcal{D}}\left[\mathcal{E}(f(x), y)\right]$.

*Proof of Proposition 1.* Consider a binary classification problem. Assume $\mathcal{P}$ be the set of possible target conditional distribution of labels given $p_s(x, y)$ and $p_t(x)$.

The forward direction is simple. If $\mathcal{P} = \{p_t(y|x)\}$ is singleton given $p_s(x, y)$ and $p_t(x)$, then the error of any classifier $f$ on the target domain is identified and is given by

$$\mathcal{E}_{\mathcal{D}^T}(f) = \mathbb{E}_{x \sim p_t(x), y \sim p_t(y|x)}\left[\mathbb{I}\left[\arg\max_{j \in \mathcal{Y}} f_j(x) \neq y\right]\right].  \tag{3}$$

For the reverse direction assume that given $p_t(x)$ and $p_s(x, y)$, we have two possible distributions $\mathcal{D}^T$ and $\mathcal{D}^{T'}$ with $p_t(y|x), p'_t(y|x) \in \mathcal{P}$ such that on some $x$ with $p_t(x) > 0$, we have $p_t(y|x) \neq p'_t(y|x)$. Consider $\mathcal{X}_M = \{x \in \mathcal{X}|p_t(x) > 0 \text{ and } p_t(y = 1|x) \neq p'_t(y = 1|x)\}$ be the set of all input covariates where the two distributions differ. We will now choose a classifier $f$ such that the error on the two distributions differ. On a subset $\mathcal{X}_M^1 = \{x \in \mathcal{X}|p_t(x) > 0 \text{ and } p_t(y = 1|x) > p'_t(y = 1|x)\}$, assume $f(x) = 0$ and on a subset $\mathcal{X}_M^2 = \{x \in \mathcal{X}|p_t(x) > 0 \text{ and } p_t(y = 1|x) < p'_t(y = 1|x)\}$, assume $f(x) = 1$. We will show that the error of $f$ on distribution with $p_t(y|x)$ is strictly greater than the error of $f$ on distribution with $p'_t(y|x)$. Formally,

$\mathcal{E}_{\mathcal{D}^T}(f) - \mathcal{E}_{\mathcal{D}^{T'}}(f)$

$= \mathbb{E}_{x \sim p_t(x), y \sim p_t(y|x)}\left[\mathbb{I}\left[\arg\max_{j \in \mathcal{Y}} f_j(x) \neq y\right]\right] - \mathbb{E}_{x \sim p_t(x), y \sim p'_t(y|x)}\left[\mathbb{I}\left[\arg\max_{j \in \mathcal{Y}} f_j(x) \neq y\right]\right]$

$= \int_{x \in \mathcal{X}_M} \mathbb{I}\left[f(x) \neq 0\right]\left(p_t(y = 0|x) - p'_t(y = 0|x)\right) p_t(x)dx$

$\quad + \int_{x \in \mathcal{X}_M} \mathbb{I}\left[f(x) \neq 1\right]\left(p_t(y = 1|x) - p'_t(y = 1|x)\right) p_t(x)dx$

$= \int_{x \in \mathcal{X}_M^2}\left(p_t(y = 0|x) - p'_t(y = 0|x)\right) p_t(x)dx + \int_{x \in \mathcal{X}_M^1}\left(p_t(y = 1|x) - p'_t(y = 1|x)\right) p_t(x)dx$

$> 0,$  (4)

where the last step follows by construction of the set $\mathcal{X}_M^1$ and $\mathcal{X}_M^2$. Since $\mathcal{E}_{\mathcal{D}^T}(f) \neq \mathcal{E}_{\mathcal{D}^{T'}}(f)$, given the information of $p_t(x)$ and $p_s(x, y)$ it is impossible to distinguish the two values of the error with classifier $f$. Thus, we obtain a contradiction on the assumption that $p_t(y|x) \neq p'_t(y|x)$. Hence, we must pose restrictions on the nature of shift such that $\mathcal{P}$ is singleton to to identify accuracy on the target. □

*Proof of Corollary 1.* The corollary follows directly from Proposition 1. Since two different target conditional distribution can lead to different error estimates without assumptions on the classifier, no method can estimate two different quantities from the same given information. We illustrate this in Example 1 next. □

# B   ESTIMATING ACCURACY IN COVARIATE SHIFT OR LABEL SHIFT

**Accuracy estimation under covariate shift assumption**   Under the assumption that $p_t(y|x) = p_s(y|x)$, accuracy on the target domain can be estimated as follows:

$$\mathcal{E}_{\mathcal{D}^T}(f) = \mathbb{E}_{(x,y) \sim \mathcal{D}^s}\left[\frac{p_t(x, y)}{p_s(x, y)}\mathbb{I}\left[f(x) \neq y\right]\right]  \tag{5}$$

$$= \mathbb{E}_{(x,y) \sim \mathcal{D}^s}\left[\frac{p_t(x)}{p_s(x)}\mathbb{I}\left[f(x) \neq y\right]\right].  \tag{6}$$

Given access to $p_t(x)$ and $p_s(x)$, one can directly estimate the expression in (6).

**Accuracy estimation under label shift assumption** Under the assumption that $p_t(x|y) = p_s(x|y)$, accuracy on the target domain can be estimated as follows:

$$\mathcal{E}_{\mathcal{D}^\mathrm{T}}(f) = \mathbb{E}_{(x,y)\sim\mathcal{D}^\mathrm{s}}\left[\frac{p_t(x,y)}{p_s(x,y)}\mathbb{I}\left[f(x) \neq y\right]\right] \tag{7}$$

$$= \mathbb{E}_{(x,y)\sim\mathcal{D}^\mathrm{s}}\left[\frac{p_t(y)}{p_s(y)}\mathbb{I}\left[f(x) \neq y\right]\right] . \tag{8}$$

Estimating importance ratios $p_t(x)/p_s(x)$ is straightforward under covariate shift assumption when the distributions $p_t(x)$ and $p_s(x)$ are known. For label shift, one can leverage moment matching approach called BBSE (Lipton et al., 2018) or likelihood minimization approach MLLS (Garg et al., 2020). Below we discuss the objective of MLLS:

$$w = \arg\max_{w\in\mathcal{W}}\mathbb{E}_{x\sim p_t(x)}\left[\log p_s(y|x)^T w\right] , \tag{9}$$

where $\mathcal{W} = \{w \mid \forall y\,, w_y \geqslant 0$ and $\sum_{y=1}^{k} w_y p_s(y) = 1\}$. MLLS objective is guaranteed to obtain consistent estimates for the importance ratios $w^*(y) = p_t(y)/p_s(y)$ under the following condition.

**Theorem 2** (Theorem 1 (Garg et al., 2020)). *If the distributions $\{p(x)|y) \,:\, y = 1,\ldots,k\}$ are strictly linearly independent, then $w^*$ is the unique maximizer of the MLLS objective* (9).

We refer interested reader to Garg et al. (2020) for details.

Above results of accuracy estimation under label shift and covariate shift can be extended to a generalized label shift and covariate shift settings. Assume a function $h : \mathcal{X} \to \mathcal{Z}$ such that $y$ is independent of $x$ given $h(x)$. In other words $h(x)$ contains all the information needed to predict label $y$. With help of $h$, we can extend estimation to following settings: (i) *Generalized covariate shift*, i.e., $p_s(y|h(x)) = p_t(y|h(x))$ and $p_s(h(x)) > 0$ for all $x \in \mathcal{X}_t$; (ii) *Generalized label shift*, i.e., $p_s(h(x)|y) = p_t(h(x)|y)$ and $p_s(y) > 0$ for all $y \in \mathcal{Y}_t$. By simply replacing $x$ with $h(x)$ in (6) and (9), we will obtain consistent error estimates under these generalized conditions.

*Proof of Example 1.* Under covariate shift using (6), we get

$$\mathcal{E}_1 = \mathbb{E}_{(x,y)\sim p_s(x,y)}\left[\frac{p_t(x)}{p_s(x)}\mathbb{I}\left[f(x) \neq y\right]\right]$$

$$= \mathbb{E}_{x\sim p_s(x,y=0)}\left[\frac{p_t(x)}{p_s(x)}\mathbb{I}\left[f(x) \neq 0\right]\right] + \mathbb{E}_{x\sim p_s(x,y=1)}\left[\frac{p_t(x)}{p_s(x)}\mathbb{I}\left[f(x) \neq 1\right]\right]$$

$$= \int \mathbb{I}\left[f(x) \neq 0\right]p_t(x)p_s(y = 0|x)dx + \int \mathbb{I}\left[f(x) \neq 1\right]p_t(x)p_s(y = 1|x)dx$$

Under label shift using (8), we get

$$\mathcal{E}_2 = \mathbb{E}_{(x,y)\sim\mathcal{D}^\mathrm{s}}\left[\frac{p_t(y)}{p_s(y)}\mathbb{I}\left[f(x) \neq y\right]\right]$$

$$= \mathbb{E}_{x\sim p_s(x,y=0)}\left[\frac{\beta}{\alpha}\mathbb{I}\left[f(x) \neq 0\right]\right] + \mathbb{E}_{x\sim p_s(x,y=1)}\left[\frac{1-\beta}{1-\alpha}\mathbb{I}\left[f(x) \neq 1\right]\right]$$

$$= \int \mathbb{I}\left[f(x) \neq 0\right]\frac{\beta}{\alpha}p_s(y = 0|x)p_s(x)dx + \int \mathbb{I}\left[f(x) \neq 1\right]\frac{(1-\beta)}{(1-\alpha)}p_s(y = 1|x)p_s(x)dx$$

Then $\mathcal{E}_1 - \mathcal{E}_2$ is given by

$$\mathcal{E}_1 - \mathcal{E}_2 = \int \mathbb{I}\left[f(x) \neq 0\right]p_s(y = 0|x)\left[p_t(x) - \frac{\beta}{\alpha}p_s(x)\right]dx$$

$$+ \int \mathbb{I}\left[f(x) \neq 1\right]p_s(y = 1|x)\left[p_t(x) - \frac{(1-\beta)}{(1-\alpha)}p_s(x)\right]dx$$

$$= \int \mathbb{I}\left[f(x) \neq 0\right]p_s(y = 0|x)\frac{(\alpha - \beta)}{\alpha}\phi(\mu_2)dx$$

$$+ \int \mathbb{I}\left[f(x) \neq 1\right]p_s(y = 1|x)\frac{(\alpha - \beta)}{1 - \alpha}\phi(\mu_1)dx . \tag{10}$$

If $\alpha > \beta$, then $\mathcal{E}_1 > \mathcal{E}_2$ and if $\alpha < \beta$, then $\mathcal{E}_1 < \mathcal{E}_2$. Since $\mathcal{E}_1 \neq \mathcal{E}_2$ for arbitrary $f$, given access to $p_s(x, y)$, and $p_t(x)$, any method that consistently estimates error under covariate shift will give an incorrect estimate under label shift and vice-versa. The reason being that the same $p_t(x)$ and $p_s(x, y)$ can correspond to error $\mathcal{E}_1$ (under covariate shift) or error $\mathcal{E}_2$ (under label shift) either of which is not discernable absent further assumptions on the nature of shift. □

## C   ALTERNATE INTERPRETATION OF ATC

Consider the following framework: Given a datum $(x, y)$, define a binary classification problem of whether the model prediction $\arg\max f(x)$ was correct or incorrect. In particular, if the model prediction matches the true label, then we assign a label 1 (positive) and conversely, if the model prediction doesn't match the true label then we assign a label 0 (negative).

Our method can be interpreted as identifying examples for correct and incorrect prediction based on the value of the score function $s(f(x))$, i.e., if the score $s(f(x))$ is greater than or equal to the threshold $t$ then our method predicts that the classifier correctly predicted datum $(x, y)$ and vice-versa if the score is less than $t$. A method that can solve this task will perfectly estimate the target performance. However, such an expectation is unrealistic. Instead, ATC expects that *most* of the examples with score above threshold are correct and most of the examples below the threshold are incorrect. More importantly, ATC selects a threshold such that the number of falsely identified correct predictions match falsely identified incorrect predictions on source distribution, thereby balancing incorrect predictions. We expect useful estimates of accuracy with ATC if the threshold transfers to target, i.e. if the number of falsely identified correct predictions match falsely identified incorrect predictions on target. This interpretation relates our method to the OOD detection literature where Hendrycks & Gimpel (2016); Hendrycks et al. (2019) highlight that classifiers tend to assign higher confidence to in-distribution examples and leverage maximum softmax confidence (or logit) to perform OOD detection.

## D   DETAILS ON THE TOY MODEL

**Skews observed in this toy model**  In Fig. 4, we illustrate the toy model used in our empirical experiment. In the same setup, we empirically observe that the margin on population with less density is large, i.e., margin is much greater than $\gamma$ when the number of observed samples is small (in Fig. 4 (d)). Building on this observation, Nagarajan et al. (2020) showed in cases when margin decreases with number of samples, a max margin classifier trained on finite samples is bound to depend on the spurious features in such cases. They referred to this skew as *geometric skew*.

Moreover, even when the number of samples are large so that we do not observe geometric skews, Nagarajan et al. (2020) showed that training for finite number of epochs, a linear classifier will have a non zero dependency on the spurious feature. They referred to this skew as *statistical skew*. Due both of these skews, we observe that a linear classifier obtained with training for finite steps on training data with finite samples, will have a non-zero dependency on the spurious feature. We refer interested reader to Nagarajan et al. (2020) for more details.

**Proof of Theorem 1**  Recall, we consider a easy-to-learn binary classification problem with two features $x = [x_{\text{inv}}, x_{\text{sp}}] \in \mathbb{R}^2$ where $x_{\text{inv}}$ is fully predictive invariant feature with a margin $\gamma > 0$ and $x_{\text{sp}} \in \{-1, 1\}$ is a spurious feature (i.e., a feature that is correlated but not predictive of the true label). Conditional on $y$, the distribution over $x_{\text{inv}}$ is given as follows:

$$x_{\text{inv}}|y \sim \begin{cases} U[\gamma, c] & y = 1 \\ U[-c, -\gamma] & y = -1 \end{cases}, \tag{11}$$

where $c$ is a fixed constant greater than $\gamma$. For simplicity, we assume that label distribution on source is uniform on $\{-1, 1\}$. $x_{\text{sp}}$ is distributed such that $P_s[x_{\text{sp}} \cdot (2y - 1) > 0] = p_{\text{sp}}$, where $p_{\text{sp}} \in (0.5, 1.0)$ controls the degree of spurious correlation. To model distribution shift, we simulate target data with different degree of spurious correlation, i.e., in target distribution $P_t[x_{\text{sp}} \cdot (2y - 1) > 0] = p'_{\text{sp}} \in [0, 1]$. Note that here we do not consider shifts in the label distribution but our result extends to arbitrary shifts in the label distribution as well.

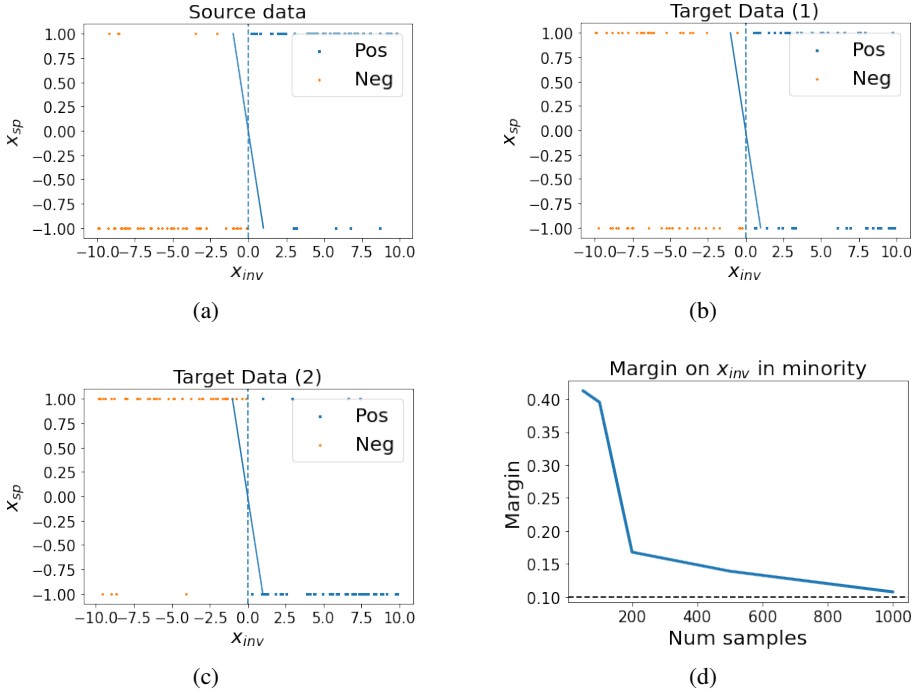

Figure 4: Illustration of toy model. (a) Source data at $n = 100$. (b) Target data with $p'_s = 0.5$. (b) Target data with $p'_s = 0.9$. (c) Margin of $x_{\text{inv}}$ in the minority group in source data. As sample size increases the margin saturates to true margin $\gamma = 0.1$.

In this setup, we examine linear sigmoid classifiers of the form $f(x) = \left[ \frac{1}{1+e^{w^T x}}, \frac{e^{w^T x}}{1+e^{w^T x}} \right]$ where $w = [w_{\text{inv}}, w_{\text{sp}}] \in \mathbb{R}^2$. We show that given a linear classifier that relies on the spurious feature and achieves a non-trivial performance on the source (i.e., $w_{\text{inv}} > 0$), ATC with maximum confidence score function *consistently* estimates the accuracy on the target distribution. Define $X_M = \{x | x_{\text{sp}} \cdot (2y - 1) < 0\}$ and $X_C = \{x | x_{\text{sp}} \cdot (2y - 1) > 0\}$. Notice that in target distributions, we are changing the fraction of examples in $X_M$ and $X_C$ but we are not changing the distribution of examples within individual set.

**Theorem 3.** *Given any classifier $f$ with $w_{inv} > 0$ in the above setting, assume that the threshold $t$ is obtained with finite sample approximation of* (1), *i.e., $t$ is selected such that*[1]

$$\sum_{i=1}^{n} \left[ \mathbb{I} \left[ \max_{j \in \mathcal{Y}} f_j(x_i) < t \right] \right] = \sum_{i=1}^{n} \left[ \mathbb{I} \left[ \arg\max_{j \in \mathcal{Y}} f_j(x_i) \neq y_i \right] \right], \quad (12)$$

*where $\{(x_i, y_i)\}_{i=1}^{n} \sim (\mathcal{D}^S)^n$ are $n$ samples from source distribution. Fix a $\delta > 0$. Assuming $n \geqslant 2 \log(4/\delta)/(1 - p_{sp})^2$, then the estimate of accuracy by ATC as in* (2) *satisfies the following with probability at least $1 - \delta$,*

$$\left| \mathbb{E}_{x \sim \mathcal{D}^T} \left[ \mathbb{I} \left[ s(f(x)) < t \right] \right] - \mathbb{E}_{(x,y) \sim \mathcal{D}^T} \left[ \mathbb{I} \left[ \arg\max_{j \in \mathcal{Y}} f_j(x) \neq y \right] \right] \right| \leqslant \sqrt{\frac{\log(8/\delta)}{n \cdot c_{sp}}}, \quad (13)$$

*where $\mathcal{D}^T$ is any target distribution considered in our setting and $c_{sp} = (1 - p_{sp})$ if $w_{sp} > 0$ and $c_{sp} = p_{sp}$ otherwise.*

---

[1]Note that this is possible because a linear classifier with sigmoid activation assigns a unique score to each point in source distribution.

*Proof.* First we consider the case of $w_{sp} > 0$. The proof follows in two simple steps. First we notice that the classifier will make an error only on some points in $X_M$ and the threshold $t$ will be selected such that the fraction of points in $X_M$ with maximum confidence less than the threshold $t$ will match the error of the classifier on $X_M$. Classifier with $w_{sp} > 0$ and $w_{inv} > 0$ will classify all the points in $X_C$ correctly. Second, since the distribution of points is not changing within $X_M$ and $X_C$, the same threshold continues to work for arbitrary shift in the fraction of examples in $X_M$, i.e., $p'_{sp}$.

Note that when $w_{sp} > 0$, the classifier makes no error on points in $X_C$ and makes an error on a subset $X_{err} = \{x | x_{sp} \cdot (2y - 1) < 0 \,\&\, (w_{inv} x_{inv} + w_{sp} x_{sp}) \cdot (2y - 1) \leqslant 0\}$ of $X_M$, i.e., $X_{err} \subseteq X_M$. Consider $X_{thres} = \{x | \arg\max_{y \in \mathcal{Y}} f_y(x) \leqslant t\}$ as the set of points that obtain a score less than or equal to $t$. Now we will show that ATC chooses a threshold $t$ such that all points in $X_C$ gets a score above $t$, i.e., $X_{thres} \subseteq X_M$. First note that the score of points close to the true separator in $X_C$, i.e., at $x_1 = (\gamma, 1)$ and $x_2 = (-\gamma, -1)$ match. In other words, score at $x_1$ matches with the score of $x_2$ by symmetricity, i.e.,

$$\arg\max_{y \in \mathcal{Y}} f_y(x_1) = \arg\max_{y \in \mathcal{Y}} f_y(x_2) = \frac{e^{w_{inv}\gamma + w_{sp}}}{(1 + e^{w_{inv}\gamma + w_{sp}})} \,. \tag{14}$$

Hence, if $t \geqslant \arg\max_{y \in \mathcal{Y}} f_y(x_1)$ then we will have $|X_{err}| < |X_{thres}|$ which is contradiction violating definition of $t$ as in (12). Thus $X_{thres} \subseteq X_M$.

Now we will relate LHS and RHS of (12) with their expectations using Hoeffdings and DKW inequality to conclude (13). Using Hoeffdings' bound, we have with probability at least $1 - \delta/4$

$$\left| \sum_{i \in X_M} \frac{\left[ \mathbb{I}\left[ \arg\max_{j \in \mathcal{Y}} f_j(x_i) \neq y_i \right] \right]}{|X_M|} - \mathbb{E}_{(x,y) \sim \mathcal{D}^{\mathrm{T}}} \left[ \mathbb{I}\left[ \arg\max_{j \in \mathcal{Y}} f_j(x) \neq y \right] \right] \right| \leqslant \sqrt{\frac{\log(8/\delta)}{2\,|X_M|}} \,. \tag{15}$$

With DKW inequality, we have with probability at least $1 - \delta/4$

$$\left| \sum_{i \in X_M} \frac{\left[ \mathbb{I}\left[ \max_{j \in \mathcal{Y}} f_j(x_i) < t' \right] \right]}{|X_M|} - \mathbb{E}_{(x,y) \sim \mathcal{D}^{\mathrm{T}}} \left[ \mathbb{I}\left[ \max_{j \in \mathcal{Y}} f_j(x) < t' \right] \right] \right| \leqslant \sqrt{\frac{\log(8/\delta)}{2\,|X_M|}} \,, \tag{16}$$

for all $t' > 0$. Combining (15) and (16) at $t' = t$ with definition (12), we have with probability at least $1 - \delta/2$

$$\left| \mathbb{E}_{x \sim \mathcal{D}^{\mathrm{T}}} \left[ \mathbb{I}\left[ s(f(x)) < t \right] \right] - \mathbb{E}_{(x,y) \sim \mathcal{D}^{\mathrm{T}}} \left[ \mathbb{I}\left[ \arg\max_{j \in \mathcal{Y}} f_j(x) \neq y \right] \right] \right| \leqslant \sqrt{\frac{\log(8/\delta)}{2\,|X_M|}} \,. \tag{17}$$

Now for the case of $w_{sp} < 0$, we can use the same arguments on $X_C$. That is, since now all the error will be on points in $X_C$ and classifier will make no error $X_M$, we can show that threshold $t$ will be selected such that the fraction of points in $X_C$ with maximum confidence less than the threshold $t$ will match the error of the classifier on $X_C$. Again, since the distribution of points is not changing within $X_M$ and $X_C$, the same threshold continues to work for arbitrary shift in the fraction of examples in $X_M$, i.e., $p'_{sp}$. Thus with similar arguments, we have

$$\left| \mathbb{E}_{x \sim \mathcal{D}^{\mathrm{T}}} \left[ \mathbb{I}\left[ s(f(x)) < t \right] \right] - \mathbb{E}_{(x,y) \sim \mathcal{D}^{\mathrm{T}}} \left[ \mathbb{I}\left[ \arg\max_{j \in \mathcal{Y}} f_j(x) \neq y \right] \right] \right| \leqslant \sqrt{\frac{\log(8/\delta)}{2\,|X_C|}} \,. \tag{18}$$

Using Hoeffdings' bound, with probability at least $1 - \delta/2$, we have

$$|X_M - n \cdot (1 - p_{sp})| \leqslant \sqrt{\frac{n \cdot \log(4/\delta)}{2}} \,. \tag{19}$$

With probability at least $1 - \delta/2$, we have

$$|X_C - n \cdot p_{sp}| \leqslant \sqrt{\frac{n \cdot \log(4/\delta)}{2}} \,. \tag{20}$$

Combining (19) and (17), we get the desired result for $w_{sp} > 0$. For $w_{sp} < 0$, we combine (20) and (18) to get the desired result. $\qquad \square$

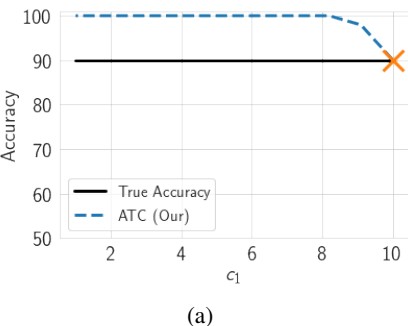

(a)

Figure 5: Failure of ATC in our toy model. Shifting the support of target class conditional $p_t(x_{\text{inv}}|y)$ may introduce a bias in ATC estimates, e.g., shrinking the support to $c_1(< c)$ (while maintaining uniform distribution) in the target leads to overestimation bias.

**Issues with IM in toy setting** As described in App. E, we observe that IM is sensitive to binning strategy. In the main paper, we include IM result with uniform mass binning with 100 bins. Empirically, we observe that we recover the true performance with IM if we use equal width binning with number of bins greater than 5.

**Biased estimation with ATC in our toy model** We assumed that both in source and target $x_{\text{inv}}|y = 1$ is uniform between $[\gamma, c]$ and $x|y = -1$ is uniform between $[-c, -\gamma]$. Shifting the support of target class conditional $p_t(x_{\text{inv}}|y)$ may introduce a bias in ATC estimates, e.g., shrinking the support to $c_1(< c)$ (while maintaining uniform distribution) in the target will lead to an over-estimation of the target performance with ATC. We show this failure in Fig. 5. The reason being that with the same threshold that we see more examples falsely identified as correct as compared to examples falsely identified as incorrect.

### D.1 A MORE GENERAL RESULT

Recall, for a given threshold $t$, we categorize an example $(x, y)$ as a falsely identified correct prediction (ficp) if the predicted label $\widehat{y} = \arg\max f(x)$ is not the same as $y$ but the predicted score $f_{\widehat{y}}(x)$ is greater than $t$. Similarly, an example is falsely identified incorrect prediction (fiip) if the predicted label $\widehat{y}$ is the same as $y$ but the predicted score $f_{\widehat{y}}(x)$ is less than $t$.

In general, we believe that our method will obtain consistent estimates in scenarios where the relative distribution of covariates doesn't change among examples that are falsely identified as incorrect and examples that are falsely identified as correct. In other words, ATC is expected to work if the distribution shift is such that falsely identified incorrect predictions match falsely identified correct prediction.

### D.2 ATC PRODUCES CONSISTENT ESTIMATE ON SOURCE DISTRIBUTION

**Proposition 2.** *Given labeled validation data $\{(x_i, y_i)\}_{i=1}^n$ from a distribution $\mathcal{D}^S$ and a model $f$, choose a threshold $t$ as in (1). Then for $\delta > 0$, with probability at least $1 - \delta$, we have*

$$\mathbb{E}_{(x,y)\sim\mathcal{D}}\left[\mathbb{I}\left[\max_{j\in\mathcal{Y}} f_j(x) < t\right] - \mathbb{I}\left[\arg\max_{j\in\mathcal{Y}} f_j(x) \neq y\right]\right] \leqslant 2\sqrt{\frac{\log(4/\delta)}{2n}} \tag{21}$$

*Proof.* The proof uses (i) Hoeffdings' inequality to relate the accuracy with expected accuracy; and (ii) DKW inequality to show the concentration of the estimated accuracy with our proposed method. Finally, we combine (i) and (ii) using the fact that at selected threshold $t$ the number of false positives is equal to the number of false negatives.

Using Hoeffdings' bound, we have with probability at least $1 - \delta/2$

$$\left|\sum_{i=1}^n\left[\mathbb{I}\left[\arg\max_{j\in\mathcal{Y}} f_j(x_i) \neq y_i\right]\right] - \mathbb{E}_{(x,y)\sim\mathcal{D}}\left[\mathbb{I}\left[\arg\max_{j\in\mathcal{Y}} f_j(x) \neq y\right]\right]\right| \leqslant \sqrt{\frac{\log(4/\delta)}{2n}}. \tag{22}$$

With DKW inequality, we have with probability at least $1 - \delta/2$

$$\left| \sum_{i=1}^{n} \left[ \mathbb{I} \left[ \max_{j \in \mathcal{Y}} f_j(x_i) < t' \right] \right] - \mathbb{E}_{(x,y) \sim \mathcal{D}} \left[ \mathbb{I} \left[ \max_{j \in \mathcal{Y}} f_j(x) < t' \right] \right] \right| \leqslant \sqrt{\frac{\log(4/\delta)}{2n}}, \qquad (23)$$

for all $t' > 0$. Finally by definition, we have

$$\sum_{i=1}^{n} \left[ \mathbb{I} \left[ \max_{j \in \mathcal{Y}} f_j(x_i) < t' \right] \right] = \sum_{i=1}^{n} \left[ \mathbb{I} \left[ \arg\max_{j \in \mathcal{Y}} f_j(x_i) \neq y_i \right] \right] \qquad (24)$$

Combining (22), (23) at $t' = t$, and (24), we have the desired result. $\qquad \square$

## E    BASLINE METHODS

**Importance-re-weighting (IM)**    If we can estimate the importance-ratios $\frac{p_t(x)}{p_s(x)}$ with just the unlabeled data from the target and validation labeled data from source, then we can estimate the accuracy as on target as follows:

$$\mathcal{E}_{\mathcal{D}^T}(f) = \mathbb{E}_{(x,y) \sim \mathcal{D}^s} \left[ \frac{p_t(x)}{p_s(x)} \mathbb{I}\left[ f(x) \neq y \right] \right]. \qquad (25)$$

As previously discussed, this is particularly useful in the setting of covariate shift (within support) where importance ratios estimation has been explored in the literature in the past. Mandolin (Chen et al., 2021) extends this approach. They estimate importance-weights with use of extra supervision about the axis along which the distribution is shifting.

In our work, we experiment with uniform mass binning and equal width binning with the number of bins in $[5, 10, 50]$. Overall, we observed that equal width binning works the best with $10$ bins. Hence throughout this paper we perform equal width binning with $10$ bins to include results with IM.

**Average Confidence (AC)**    If we expect the classifier to be argmax calibrated on the target then average confidence is equal to accuracy of the classifier. Formally, by definition of argmax calibration of $f$ on any distribution $\mathcal{D}$, we have

$$\mathcal{E}_{\mathcal{D}}(f) = \mathbb{E}_{(x,y) \sim \mathcal{D}} \left[ \mathbb{I} \left[ y \notin \arg\max_{j \in \mathcal{Y}} f_j(x) \right] \right] = \mathbb{E}_{(x,y) \sim \mathcal{D}} \left[ \max_{j \in \mathcal{Y}} f_j(x) \right]. \qquad (26)$$

**Difference Of Confidence**    We estimate the error on target by subtracting difference of confidences on source and target (as a distributional distance (Guillory et al., 2021)) from expected error on source distribution, i.e, $\text{DOC}_{\mathcal{D}^T} = \mathbb{E}_{x \sim \mathcal{D}^s} \left[ \mathbb{I} \left[ \arg\max_{j \in \mathcal{Y}} f_j(x) \neq y \right] \right] + \mathbb{E}_{x \sim \mathcal{D}^T} \left[ \max_{j \in \mathcal{Y}} f_j(x) \right] - \mathbb{E}_{x \sim \mathcal{D}^s} \left[ \max_{j \in \mathcal{Y}} f_j(x) \right]$. This is referred to as DOC-Feat in (Guillory et al., 2021).

**Generalized Disagreement Equality (GDE)**    Jiang et al. (2021) proposed average disagreement of two models (trained on the same training set but with different initialization and/or different data ordering) as a approximate measure of accuracy on the underlying data, i.e.,

$$\mathcal{E}_{\mathcal{D}}(f) = \mathbb{E}_{(x,y) \sim \mathcal{D}} \left[ \mathbb{I} \left[ f(x) \neq f'(x) \right] \right]. \qquad (27)$$

They show that marginal calibration of the model is sufficient to have expected test error equal to the expected of average disagreement of two models where the latter expectation is also taken over the models used to calculate disagreement.

## F    DETAILS ON THE DATASET SETUP

In our empirical evaluation, we consider both natural and synthetic distribution shifts. We consider shifts on ImageNet (Russakovsky et al., 2015), CIFAR Krizhevsky & Hinton (2009), FMoW-WILDS (Christie et al., 2018), RxRx1-WILDS (Taylor et al., 2019), Amazon-WILDS (Ni et al., 2019), CivilComments-WILDS (Borkan et al., 2019), and MNIST LeCun et al. (1998) datasets.

| Train (Source) | Valid (Source) | Evaluation (Target) |
|---|---|---|
| MNIST (train) | MNIST (valid) | USPS, SVHN and Q-MNIST |
| CIFAR10 (train) | CIFAR10 (valid) | CIFAR10v2, 95 CIFAR10-C datasets (Fog and Motion blur, etc. ) |
| CIFAR100 (train) | CIFAR100 (valid) | 95 CIFAR100-C datasets (Fog and Motion blur, etc. ) |
| FMoW (2002-12) (train) | FMoW (2002-12) (valid) | FMoW $\{(2013\text{-}15, 2016\text{-}17) \times$ (All, Africa, Americas, Oceania, Asia, and Europe)$\}$ |
| RxRx1 (train) | RxRx1(id-val) | RxRx1 (id-test, OOD-val, OOD-test) |
| Amazon (train) | Amazon (id-val) | Amazon (OOD-val, OOD-test) |
| CivilComments (train) | CivilComments (id-val) | CiviComments (8 demographic identities male, female, LGBTQ, Christian, Muslim, other religions, Black, and White) |
| ImageNet (train) | ImageNet (valid) | 3 ImageNetv2 datasets, ImageNet-Sketch, 95 ImageNet-C datasets |
| ImageNet-200 (train) | ImageNet-200 (valid) | 3 ImageNet-200v2 datasets, ImageNet-R, ImageNet200-Sketch, 95 ImageNet200-C datasets |
| BREEDS (train) | BREEDS (valid) | Same subpopulations as train but unseen images from natural and synthetic shifts in ImageNet, Novel subpopulations on natural and synthetic shifts |

Table 2: Details of the test datasets considered in our evaluation.

*ImageNet setup.* First, we consider synthetic shifts induced to simulate 19 different visual corruptions (e.g., shot noise, motion blur, pixelation etc.) each with 5 different intensities giving us a total of 95 datasets under ImageNet-C (Hendrycks & Dietterich, 2019). Next, we consider natural distribution shifts due to differences in the data collection process. In particular, we consider 3 ImageNetv2 (Recht et al., 2019) datasets each using a different strategy to collect test sets. We also evaluate performance on images with artistic renditions of object classes, i.e., ImageNet-R (Hendrycks et al., 2021) and ImageNet-Sketch (Wang et al., 2019) with hand drawn sketch images. Note that renditions dataset only contains 200 classes from ImageNet. Hence, in the main paper we include results on ImageNet restricted to these 200 classes, which we call as ImageNet-200, and relegate results on ImageNet with 1k classes to appendix.

We also consider BREEDS benchmark (Santurkar et al., 2020) in our evaluation to assess robustness to subpopulation shifts, in particular, to understand how accuracy estimation methods behave when novel subpopulations not observed during training are introduced. BREEDS leverages class hierarchy in ImageNet to repurpose original classes to be the subpopulations and defines a classification task on superclasses. Subpopulation shift is induced by directly making the subpopulations present in the training and test distributions disjoint. Overall, BREEDS benchmark contains 4 datasets ENTITY-13, ENTITY-30, LIVING-17, NON-LIVING-26, each focusing on different subtrees in the hierarchy. To generate BREEDS dataset on top of ImageNet, we use the open source library: `https://github.com/MadryLab/BREEDS-Benchmarks`. We focus on natural and synthetic shifts as in ImageNet on same and different subpopulations in BREEDs. Thus for both the subpopulation (same or novel), we obtain a total of 99 target datasets.

*CIFAR setup.* Similar to the ImageNet setup, we consider (i) synthetic shifts (CIFAR-10-C) due to common corruptions; and (ii) natural distribution shift (i.e., CIFARv2 (Recht et al., 2018; Torralba et al., 2008)) due to differences in data collection strategy on on CIFAR-10 (Krizhevsky & Hinton, 2009). On CIFAR-100, we just have synthetic shifts due to common corruptions.

*FMoW-WILDS setup.* In order to consider distribution shifts faced in the wild, we consider FMoW-WILDS (Koh et al., 2021; Christie et al., 2018) from WILDS benchmark, which contains satellite images taken in different geographical regions and at different times. We obtain 12 different OOD target sets by considering images between years 2013–2016 and 2016–2018 and by considering five geographical regions as subpopulations (Africa, Americas, Oceania, Asia, and Europe) separately and together.

*RxRx1–WILDS setup.* Similar to FMoW, we consider RxRx1-WILDS (Taylor et al., 2019) from WILDS benchmark, which contains image of cells obtained by fluorescent microscopy and the task

is to genetic treatments the cells received. We obtain 3 target datasets with shift induced by batch effects which make it difficult to draw conclusions from data across experimental batches.

*Amazon-*WILDS *setup.* For natural language task, we consider Amazon-WILDS (Ni et al., 2019) dataset from WILDS benchmark, which contains review text and the task is get a corresponding star rating from 1 to 5. We obtain 2 target datasets by considered shifts induced due to different set of reviewers than the training set.

*CivilComments-*WILDS *setup.* We also consider CivilComments-WILDS (Borkan et al., 2019) from WILDS benchmark, which contains text comments and the task is to classify them for toxicity. We obtain 18 target datasets depending on whether a comment mentions each of the 8 demographic identities male, female, LGBTQ, Christian, Muslim, other religions, Black, and White.

*MNIST setup.* For completeness, we also consider distribution shifts on MNIST (LeCun et al., 1998) digit classification as in the prior work (Deng & Zheng, 2021). We use three real shifted datasets, i.e., USPS (Hull, 1994), SVHN (Netzer et al., 2011) and QMNIST (Yadav & Bottou, 2019).

## G    DETAILS ON THE EXPERIMENTAL SETUP

All experiments were run on NVIDIA Tesla V100 GPUs. We used PyTorch (Paszke et al., 2019) for experiments.

**Deep nets**  We consider a 4-layered MLP. The PyTorch code for 4-layer MLP is as follows:

```
nn.Sequential(nn.Flatten(),
    nn.Linear(input_dim, 5000, bias=True),
    nn.ReLU(),
    nn.Linear(5000, 5000, bias=True),
    nn.ReLU(),
    nn.Linear(5000, 50, bias=True),
    nn.ReLU(),
    nn.Linear(50, num_label, bias=True)
    )
```

We mainly experiment convolutional nets. In particular, we use ResNet18 (He et al., 2016), ResNet50, and DenseNet121 (Huang et al., 2017) architectures with their default implementation in PyTorch. Whenever we initial our models with pre-trained models, we again use default models in PyTorch.

**Hyperparameters and Training details**  As mentioned in the main text we do not alter the standard training procedures and hyperparameters for each task. We present results at final model, however, we observed that the same results extend to an early stopped model as well. For completeness, we include these details below:

*CIFAR10 and CIFAR100*  We train DenseNet121 and ResNet18 architectures from scratch. We use SGD training with momentum of 0.9 for 300 epochs. We start with learning rate 0.1 and decay it by multiplying it with 0.1 every 100 epochs. We use a weight decay of $5^-4$. We use batch size of 200. For CIFAR10, we also experiment with the same models pre-trained on ImageNet.

*ImageNet*  For training, we use Adam with a batch size of 64 and learning rate 0.0001. Due to huge size of ImageNet, we could only train two models needed for GDE for 10 epochs. Hence, for relatively small scale experiments, we also perform experiments on ImageNet subset with 200 classes, which we call as ImageNet-200 with the same training procedure. These 200 classes are the same classes as in ImageNet-R dataset. This not only allows us to train ImageNet for 50 epochs but also allows us to use ImageNet-R in our testbed. On the both the datasets, we observe a similar superiority with ATC. Note that all the models trained here were initialized with a pre-trained ImageNet model with the last layer replaced with random weights.

*FMoW-*WILDS    For all experiments, we follow Koh et al. (2021) and use two architectures DenseNet121 and ResNet50, both pre-trained on ImageNet. We use the Adam optimizer (Kingma & Ba, 2014) with an initial learning rate of $10^{-4}$ that decays by 0.96 per epoch, and train for 50 epochs and with a batch size of 64.

*RxRx1*-WILDS    For all experiments, we follow Koh et al. (2021) and use two architectures DenseNet121 and ResNet50, both pre-trained on ImageNet. We use Adam optimizer with a learning rate of $1e-4$ and L2-regularization strength of $1e-5$ with a batch size of 75 for 90 epochs. We linearly increase the learning rate for 10 epochs, then decreasing it following a cosine learning rate schedule. Finally, we pick the model that obtains highest in-distribution validation accuracy.

*Amazon*-WILDS    For all experiments, we follow Koh et al. (2021) and finetuned DistilBERT-base-uncased models (Sanh et al., 2019), using the implementation from Wolf et al. (2020), and with the following hyperparameter settings: batch size 8; learning rate $1e-5$ with the AdamW optimizer (Loshchilov & Hutter, 2017); L2-regularization strength 0.01; 3 epochs with early stopping; and a maximum number of tokens of 512.

*CivilComments*-WILDS    For all experiments, we follow Koh et al. (2021) and fine-tuned DistilBERT-base-uncased models (Sanh et al., 2019), using the implementation from Wolf et al. (2020) and with the following hyperparameter settings: batch size 16; learning rate $1e-5$ with the AdamW optimizer (Loshchilov & Hutter, 2017) for 5 epochs; L2-regularization strength 0.01; and a maximum number of tokens of 300.

*Living17 and Nonliving26 from* BREEDS    For training, we use SGD with a batch size of 128, weight decay of $10^{-4}$, and learning rate 0.1. Models were trained until convergence. Models were trained for a total of 450 epochs, with 10-fold learning rate drops every 150 epochs. Note that since we want to evaluate models for novel subpopulations no pre-training was used. We train two architectures DenseNet121 and ResNet50.

*Entity13 and Entity30 from* BREEDS    For training, we use SGD with a batch size of 128, weight decay of $10^{-4}$, and learning rate 0.1. Models were trained until convergence. Models were trained for a total of 300 epochs, with 10-fold learning rate drops every 100 epochs. Note that since we want to evaluate models for novel subpopulations no pre-training was used. We train two architectures DenseNet121 and ResNet50.

*MNIST*    For MNIST, we train a MLP described above with SGD with momentum 0.9 and learning rate 0.01 for 50 epochs. We use weight decay of $10^{-5}$ and batch size as 200.

We have a single number for CivilComments because it is a binary classification task. For multiclass problems, ATC-NE and ATC-MC can lead to different ordering of examples when ranked with the corresponding scoring function. Temperature scaling on top can further alter the ordering of examples. The changed ordering of examples yields different thresholds and different accuracy estimates. However for binary classification, the two scoring functions are the same as entropy (i.e. $p\log(p) + (1-p)\log(p)$) has a one-to-one mapping to the max conf for $p \in [0, 1]$. Moreover, temperature scaling also doesn't change the order of points for binary classification problems. Hence for the binary classification problems, both the scoring functions with and without temperature scaling yield the same estimates. We have made this clear in the updated draft.

**Implementation for Temperature Scaling**    We use temperature scaling implementation from `https://github.com/kundajelab/abstention`. We use validation set (the same we use to obtain ATC threshold or DOC source error estimate) to tune a single temperature parameter.

## G.1    DETAILS ON FIG. 1 (RIGHT) SETUP

For vision datasets, we train a DenseNet model with the exception of FCN model for MNIST dataset. For language datasets, we fine-tune a DistilBERT-base-uncased model. For each of these models, we use the exact same setup as described Sec. G. Importantly, to obtain errors on the same scale, we rescale all the errors by subtracting the error of Average Confidence method for each model. Results are reported as mean of the re-scaled errors over 4 seeds.

# H  SUPPLEMENTARY RESULTS

## H.1  CIFAR PRETRAINING ABLATION

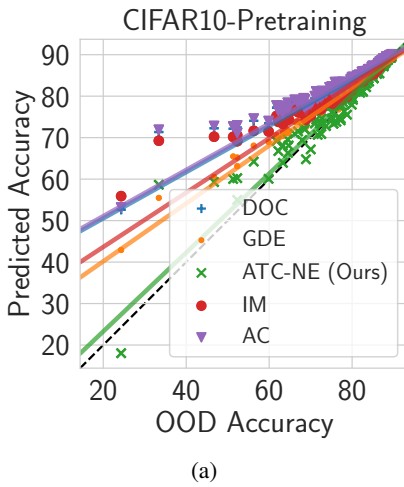

(a)

Figure 6: Results with a pretrained DenseNet121 model on CIFAR10. We observe similar behaviour as that with a model trained from scratch.

## H.2  BREEDS RESULTS WITH REGRESSION MODEL

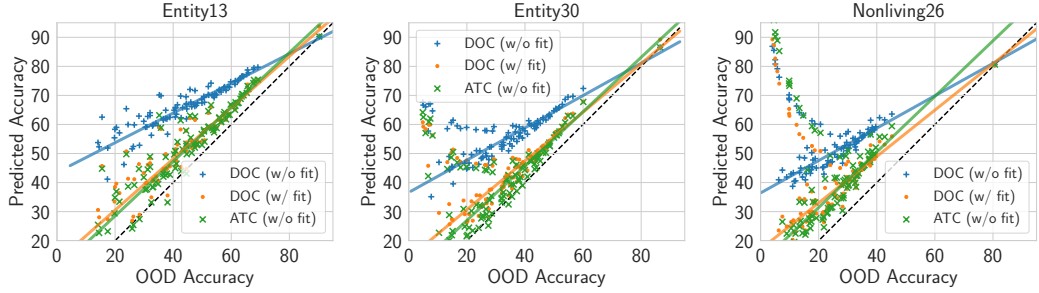

Figure 7: Scatter plots for DOC with linear fit. Results parallel to Fig. 3(Middle) on other BREEDS dataset.

| Dataset | DOC (w/o fit) | DOC (w fit) | ATC-MC (Ours) (w/o fit) |
|---|---|---|---|
| LIVING-17 | 24.32 | 13.65 | **10.07** |
| NONLIVING-26 | 29.91 | **18.13** | 19.37 |
| ENTITY-13 | 22.18 | 8.63 | 8.01 |
| ENTITY-30 | 24.71 | 12.28 | **10.21** |

Table 5: *Mean Absolute estimation Error (MAE) results for BREEDs datasets with novel populations in our setup.* After fitting a robust linear model for DOC on same subpopulation, we show predicted accuracy on different subpopulations with fine-tuned DOC (i.e., DOC (w/ fit)) and compare with ATC without any regression model, i.e., ATC (w/o fit). While observe substantial improvements in MAE from DOC (w/o fit) to DOC (w/ fit), ATC (w/o fit) continues to outperform even DOC (w/ fit).

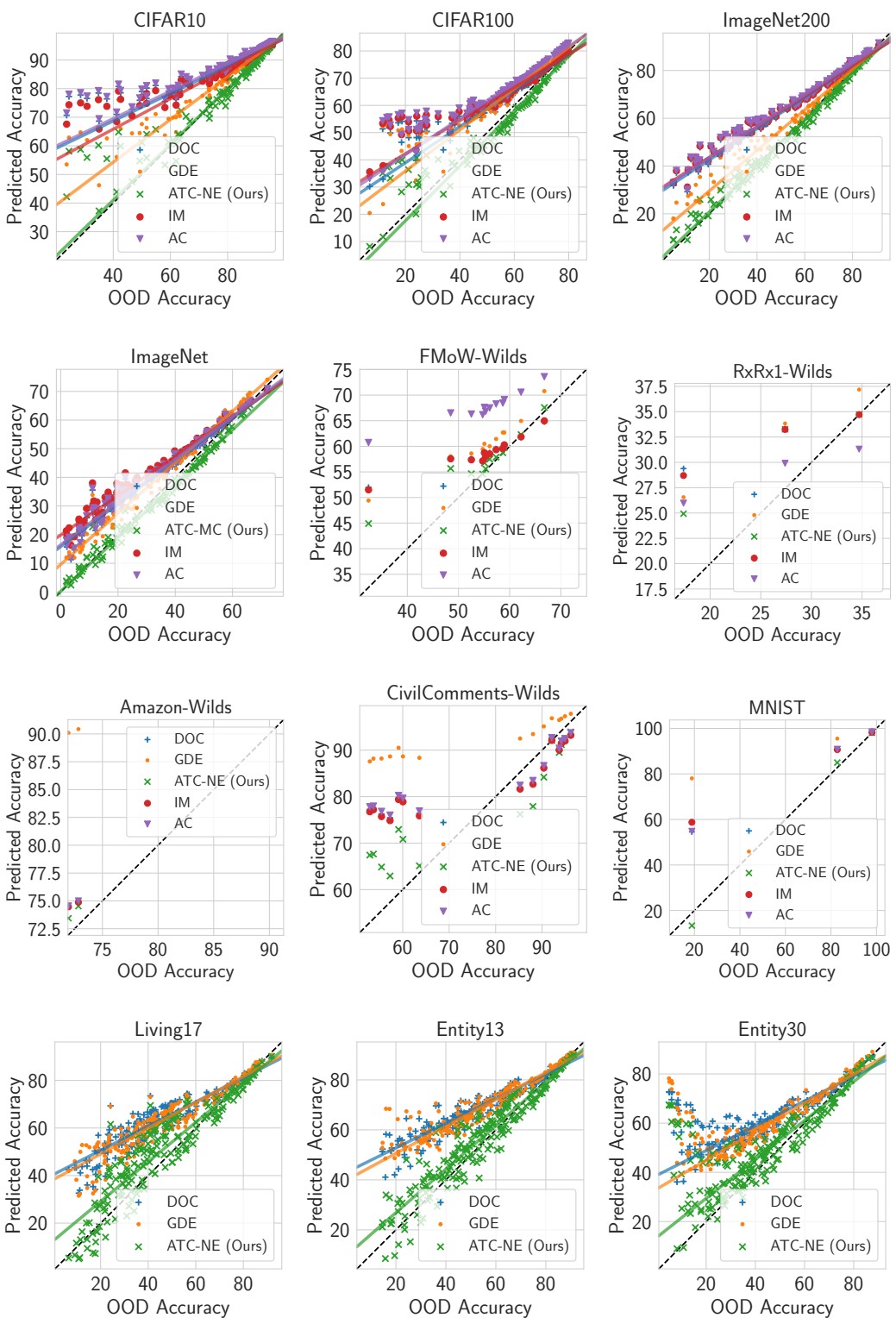

Figure 8: Scatter plot of predicted accuracy versus (true) OOD accuracy. For vision datasets except MNIST we use a DenseNet121 model. For MNIST, we use a FCN. For language datasets, we use DistillBert-base-uncased. Results reported by aggregating accuracy numbers over 4 different seeds.

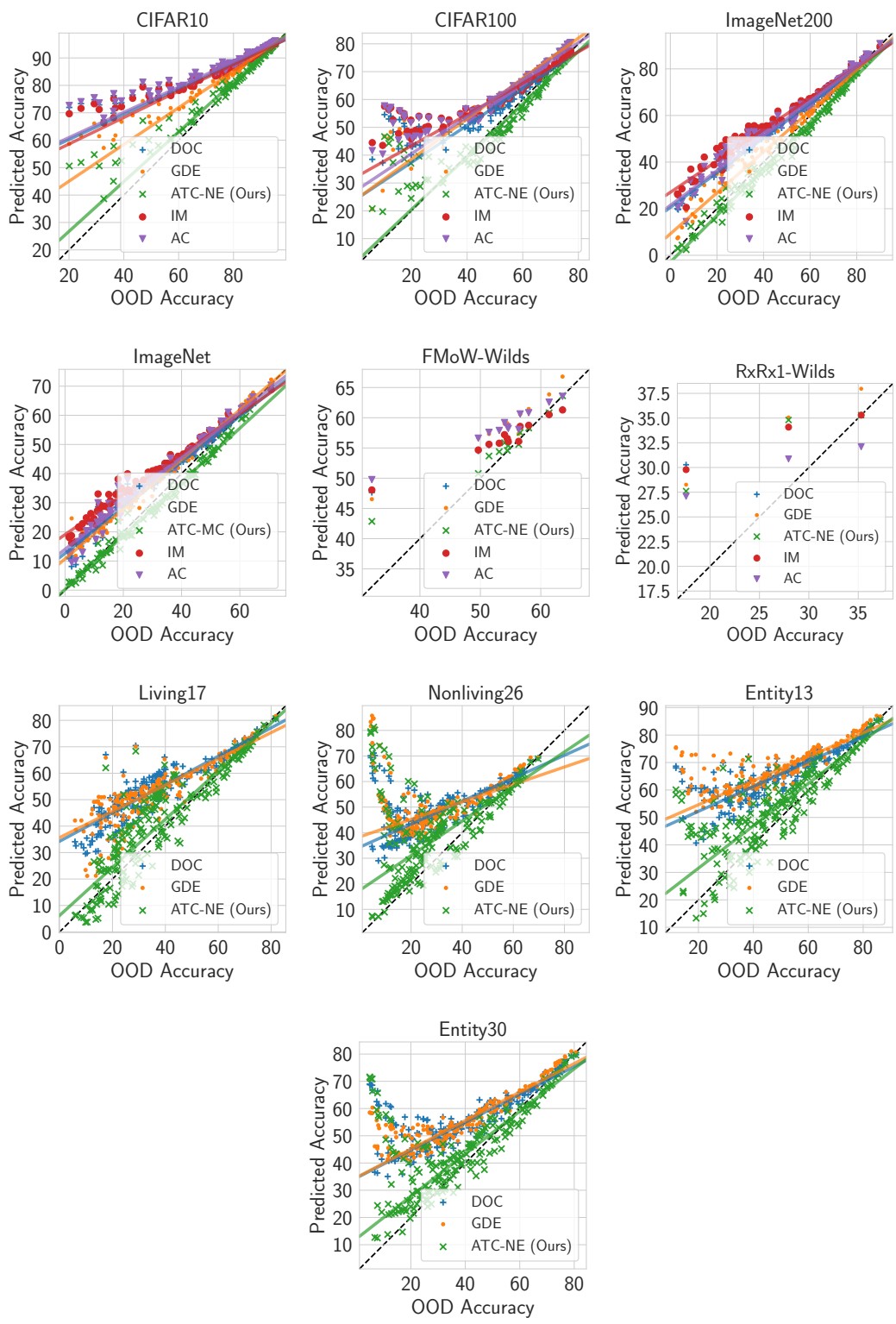

Figure 9: Scatter plot of predicted accuracy versus (true) OOD accuracy for vision datasets except MNIST with a ResNet50 model. Results reported by aggregating MAE numbers over 4 different seeds.

| Dataset | Shift | IM Pre T | IM Post T | AC Pre T | AC Post T | DOC Pre T | DOC Post T | GDE Post T | ATC-MC (Ours) Pre T | ATC-MC (Ours) Post T | ATC-NE (Ours) Pre T | ATC-NE (Ours) Post T |
|---|---|---|---|---|---|---|---|---|---|---|---|---|
| CIFAR10 | Natural | 6.60 (0.35) | 5.74 (0.30) | 9.88 (0.16) | 6.89 (0.13) | 7.25 (0.15) | 6.07 (0.16) | 4.77 (0.13) | 3.21 (0.49) | 3.02 (0.40) | 2.99 (0.37) | **2.85** (0.29) |
| CIFAR10 | Synthetic | 12.33 (0.51) | 10.20 (0.48) | 16.50 (0.26) | 11.91 (0.17) | 13.87 (0.18) | 11.08 (0.17) | 6.55 (0.35) | 4.65 (0.55) | 4.25 (0.55) | 4.21 (0.55) | **3.87** (0.75) |
| CIFAR100 | Synthetic | 13.69 (0.55) | 11.51 (0.41) | 23.61 (1.16) | 13.10 (0.80) | 14.60 (0.77) | 10.14 (0.64) | 9.85 (0.57) | 5.50 (0.70) | **4.75** (0.73) | 4.72 (0.74) | 4.94 (0.74) |
| ImageNet200 | Natural | 12.37 (0.25) | 8.19 (0.33) | 22.07 (0.08) | 8.61 (0.25) | 15.17 (0.11) | 7.81 (0.29) | 5.13 (0.08) | 4.37 (0.39) | 2.04 (0.24) | 3.79 (0.30) | **1.45** (0.27) |
| ImageNet200 | Synthetic | 19.86 (1.38) | 12.94 (1.81) | 32.44 (1.00) | 13.35 (1.30) | 25.02 (1.10) | 12.38 (1.38) | 5.41 (0.89) | 5.93 (1.38) | 3.09 (0.87) | 5.00 (1.28) | **2.68** (0.45) |
| ImageNet | Natural | 7.77 (0.27) | 6.50 (0.33) | 18.13 (0.23) | 6.02 (0.34) | 8.13 (0.27) | 5.76 (0.37) | 6.23 (0.41) | 3.88 (0.53) | 2.17 (0.62) | 2.06 (0.54) | **0.80** (0.44) |
| ImageNet | Synthetic | 13.39 (0.53) | 10.12 (0.63) | 24.62 (0.64) | 8.51 (0.71) | 13.55 (0.61) | 7.90 (0.72) | 6.32 (0.33) | 3.34 (0.53) | **2.53** (0.36) | 2.61 (0.33) | 4.89 (0.83) |
| FMoW-WILDS | Natural | 5.53 (0.33) | 4.31 (0.63) | 33.53 (0.13) | 12.84 (12.06) | 5.94 (0.36) | 4.45 (0.77) | 5.74 (0.55) | 3.06 (0.36) | **2.70** (0.54) | 3.02 (0.35) | 2.72 (0.44) |
| RxRx1-WILDS | Natural | 5.80 (0.17) | 5.72 (0.15) | 7.90 (0.24) | 4.84 (0.09) | 5.98 (0.15) | 5.98 (0.13) | 6.03 (0.08) | 4.66 (0.38) | **4.56** (0.38) | 4.41 (0.31) | 4.47 (0.26) |
| Amazon-WILDS | Natural | 2.40 (0.08) | 2.29 (0.09) | 8.01 (0.53) | 2.38 (0.17) | 2.40 (0.09) | 2.28 (0.09) | 17.87 (0.18) | 1.65 (0.06) | **1.62** (0.05) | 1.60 (0.14) | 1.59 (0.15) |
| CivilCom.-WILDS | Natural | 12.64 (0.52) | 10.80 (0.48) | 16.76 (0.53) | 11.03 (0.49) | 13.31 (0.52) | 10.99 (0.49) | 16.65 (0.25) | | **7.14** (0.41) | | |
| MNIST | Natural | 18.48 (0.45) | 15.99 (1.53) | 21.17 (0.24) | 14.81 (3.89) | 20.19 (0.23) | 14.56 (3.47) | 24.42 (0.41) | 5.02 (0.44) | **2.40** (1.83) | 3.14 (0.49) | 3.50 (0.17) |
| ENTITY-13 | Same | 16.23 (0.77) | 11.14 (0.65) | 24.97 (0.70) | 10.88 (0.77) | 19.08 (0.65) | 10.47 (0.72) | 10.71 (0.74) | 5.39 (0.92) | **3.88** (0.61) | 4.58 (0.85) | 4.19 (0.16) |
| ENTITY-13 | Novel | 28.53 (0.82) | 22.02 (0.68) | 38.33 (0.75) | 21.64 (0.86) | 32.43 (0.69) | 21.22 (0.80) | 20.61 (0.60) | 13.58 (1.15) | 10.28 (1.34) | 12.25 (1.21) | **6.63** (0.93) |
| ENTITY-30 | Same | 18.59 (0.51) | 14.46 (0.52) | 28.82 (0.43) | 14.30 (0.71) | 21.63 (0.37) | 13.46 (0.59) | 12.92 (0.14) | 9.12 (0.62) | **7.75** (0.72) | 8.15 (0.68) | 7.64 (0.88) |
| ENTITY-30 | Novel | 32.34 (0.60) | 26.85 (0.58) | 44.02 (0.56) | 26.27 (0.79) | 36.82 (0.47) | 25.42 (0.68) | 23.16 (0.12) | 17.75 (0.76) | 14.30 (0.85) | 15.60 (0.86) | **10.57** (0.86) |
| NONLIVING-26 | Same | 18.66 (0.76) | 17.17 (0.74) | 26.39 (0.82) | 16.14 (0.81) | 19.86 (0.67) | 15.58 (0.76) | 16.63 (0.45) | 10.87 (0.98) | **10.24** (0.83) | 10.07 (0.92) | 10.26 (1.18) |
| NONLIVING-26 | Novel | 33.43 (0.67) | 31.53 (0.65) | 41.66 (0.67) | 29.87 (0.71) | 35.13 (0.54) | 29.31 (0.64) | 29.56 (0.21) | 21.70 (0.86) | 20.12 (0.75) | 19.08 (0.82) | **18.26** (1.12) |
| LIVING-17 | Same | 12.63 (1.25) | 11.05 (1.20) | 18.32 (1.01) | 10.46 (1.12) | 14.43 (1.11) | 10.14 (1.16) | 9.87 (0.61) | 4.57 (0.71) | **3.95** (0.48) | 3.81 (0.22) | 4.21 (0.53) |
| LIVING-17 | Novel | 29.03 (1.44) | 26.96 (1.38) | 35.67 (1.09) | 26.11 (1.27) | 31.73 (1.19) | 25.73 (1.35) | 23.53 (0.52) | 16.15 (1.36) | 14.49 (1.46) | 12.97 (1.52) | **11.39** (1.72) |

Table 3: *Mean Absolute estimation Error (MAE) results for different datasets in our setup grouped by the nature of shift.* 'Same' refers to same subpopulation shifts and 'Novel' refers novel subpopulation shifts. We include details about the target sets considered in each shift in Table 2. Post T denotes use of TS calibration on source. For language datasets, we use DistilBERT-base-uncased, for vision dataset we report results with DenseNet model with the exception of MNIST where we use FCN. Across all datasets, we observe that ATC achieves superior performance (lower MAE is better). For GDE post T and pre T estimates match since TS doesn't alter the argmax prediction. Results reported by aggregating MAE numbers over 4 different seeds. Values in parenthesis (i.e., (·)) denote standard deviation values.

| Dataset | Shift | IM | | AC | | DOC | | GDE | ATC-MC (Ours) | | ATC-NE (Ours) | |
|---|---|---|---|---|---|---|---|---|---|---|---|---|
| | | Pre T | Post T | Pre T | Post T | Pre T | Post T | Post T | Pre T | Post T | Pre T | Post T |
| CIFAR10 | Natural | 7.14 (0.14) | 6.20 (0.11) | 10.25 (0.31) | 7.06 (0.33) | 7.68 (0.28) | 6.35 (0.27) | 5.74 (0.25) | 4.02 (0.38) | 3.85 (0.30) | 3.76 (0.33) | **3.38** (0.32) |
| | Synthetic | 12.62 (0.76) | 10.75 (0.71) | 16.50 (0.28) | 11.91 (0.24) | 13.93 (0.29) | 11.20 (0.28) | 7.97 (0.13) | 5.66 (0.64) | 5.03 (0.71) | 4.87 (0.71) | **3.63** (0.62) |
| CIFAR100 | Synthetic | 12.77 (0.43) | 12.34 (0.68) | 16.89 (0.20) | 12.73 (2.59) | 11.18 (0.35) | 9.63 (1.25) | 12.00 (0.48) | 5.61 (0.51) | **5.55** (0.55) | 5.65 (0.35) | 5.76 (0.27) |
| ImageNet200 | Natural | 12.63 (0.59) | 7.99 (0.47) | 23.08 (0.31) | 7.22 (0.22) | 15.40 (0.42) | 6.33 (0.24) | 5.00 (0.36) | 4.60 (0.63) | 1.80 (0.17) | 4.06 (0.69) | **1.38** (0.29) |
| | Synthetic | 20.17 (0.74) | 11.74 (0.80) | 33.69 (0.73) | 9.51 (0.51) | 25.49 (0.66) | 8.61 (0.50) | 4.19 (0.14) | 5.37 (0.88) | 2.78 (0.23) | 4.53 (0.79) | 3.58 (0.33) |
| ImageNet | Natural | 8.09 (0.25) | 6.42 (0.28) | 21.66 (0.38) | 5.91 (0.22) | 8.53 (0.26) | 5.21 (0.25) | 5.90 (0.44) | 3.93 (0.26) | 1.89 (0.21) | 2.45 (0.16) | **0.73** (0.10) |
| | Synthetic | 13.93 (0.14) | 9.90 (0.23) | 28.05 (0.39) | 7.56 (0.13) | 13.82 (0.31) | 6.19 (0.07) | 6.70 (0.52) | 3.33 (0.25) | 2.55 (0.25) | 2.12 (0.31) | 5.06 (0.27) |
| FMoW-WILDS | Natural | 5.15 (0.19) | 3.55 (0.41) | 34.64 (0.22) | 5.03 (0.29) | 5.58 (0.17) | 3.46 (0.37) | 5.08 (0.46) | 2.59 (0.32) | 2.33 (0.28) | 2.52 (0.25) | **2.22** (0.30) |
| RxRx1-WILDS | Natural | 6.17 (0.20) | 6.11 (0.24) | 21.05 (0.31) | **5.21** (0.18) | 6.54 (0.21) | 6.27 (0.20) | 6.82 (0.31) | 5.30 (0.30) | **5.20** (0.44) | **5.19** (0.43) | 5.63 (0.55) |
| ENTITY-13 | Same | 18.32 (0.29) | 14.38 (0.53) | 27.79 (1.18) | 13.56 (0.58) | 20.50 (0.47) | 13.22 (0.58) | 16.09 (0.84) | 9.35 (0.79) | 7.50 (0.65) | 7.80 (0.62) | **6.94** (0.71) |
| | Novel | 28.82 (0.30) | 24.03 (0.55) | 38.97 (1.32) | 22.96 (0.59) | 31.66 (0.54) | 22.61 (0.58) | 25.26 (1.08) | 17.11 (0.84) | 13.96 (0.93) | 14.75 (0.64) | **9.94** (0.78) |
| ENTITY-30 | Same | 16.91 (1.33) | 14.61 (1.11) | 26.84 (2.15) | 14.37 (1.34) | 18.60 (1.69) | 13.11 (1.30) | 13.74 (1.07) | 8.54 (1.47) | 7.94 (1.38) | **7.77** (1.44) | 8.04 (1.51) |
| | Novel | 28.66 (1.16) | 25.83 (0.88) | 39.21 (2.03) | 25.03 (1.11) | 30.95 (1.64) | 23.73 (1.11) | 23.15 (0.51) | 15.57 (1.44) | 13.24 (1.15) | 12.44 (1.26) | **11.05** (1.13) |
| NONLIVING-26 | Same | 17.43 (0.90) | 15.95 (0.86) | 27.70 (0.90) | 15.40 (0.69) | 18.06 (1.00) | 14.58 (0.78) | 16.99 (1.25) | 10.79 (0.62) | **10.13** (0.32) | 10.05 (0.46) | 10.29 (0.79) |
| | Novel | 29.51 (0.86) | 27.75 (0.82) | 40.02 (0.76) | 26.77 (0.82) | 30.36 (0.95) | 25.93 (0.80) | 27.70 (1.42) | 19.64 (0.68) | 17.75 (0.53) | 16.90 (0.60) | **15.69** (0.83) |
| LIVING-17 | Same | 14.28 (0.96) | 12.21 (0.93) | 23.46 (1.16) | 11.16 (0.90) | 15.22 (0.96) | 10.78 (0.99) | 10.49 (0.97) | 4.92 (0.57) | **4.23** (0.42) | 4.19 (0.35) | 4.73 (0.24) |
| | Novel | 28.91 (0.66) | 26.35 (0.73) | 38.62 (1.01) | 24.91 (0.61) | 30.32 (0.59) | 24.52 (0.74) | 22.49 (0.85) | 15.42 (0.59) | 13.02 (0.53) | 12.29 (0.73) | **10.34** (0.62) |

Table 4: *Mean Absolute estimation Error (MAE) results for different datasets in our setup grouped by the nature of shift for ResNet model.* 'Same' refers to same subpopulation shifts and 'Novel' refers novel subpopulation shifts. We include details about the target sets considered in each shift in Table 2. Post T denotes use of TS calibration on source. Across all datasets, we observe that ATC achieves superior performance (lower MAE is better). For GDE post T and pre T estimates match since TS doesn't alter the argmax prediction. Results reported by aggregating MAE numbers over 4 different seeds. Values in parenthesis (i.e., $(\cdot)$) denote standard deviation values.

