# OpenReview forum: "Leveraging unlabeled data to predict out-of-distribution performance"
_ICLR.cc/2022/Conference — ICLR 2022 Poster_

### Official Review · Reviewer_zA7T · 2021-10-24

**Correctness:** 4
**Technical Novelty And Significance:** 3
**Empirical Novelty And Significance:** 3
**Recommendation:** 6
**Confidence:** 3

**Main Review:**

The problem of predicting accuracy of the models is interesting and novel to me. How to identify the accuracy of a model under distribution shift is a practical challenge. The proposed method of ATC is extremely simple and makes intuitive sense. It is also reasonable to claim that identifying the accuracy is as hard as identifying the optimal predictor, and the efficacy of any method depends on the assumptions on the nature of the shift.

Quality: The submission is technically sound. The claims in the contribution are well-supported by theoretical analyses and empirically results. It is a complete piece of work that outperforms prior work empirically.

Clarity: This paper is well-written and easy to follow. The problem is also well-motivated. The experimental details are also very specific, such that reproducing the results should be possible.

**Summary Of The Paper:**

This paper studies how to predict the target domain accuracy using only labeled source data and unlabeled target data. They propose Average Thresholded Confidence (ATC) to learn a threshold on the model's confidence. In particular, ATC predicts accuracy as the fraction of unlabeled examples for which model confidence exceeds the threshold. Extensive experiments have been conducted to show that ATC outperforms prior work across multiple model architectures, types of distribution shifts, and datasets. Theoretically, they prove that identifying the accuracy is as hard as identifying the optimal predictor, and the efficacy of any method depends on the assumptions on the nature of the shift.

**Summary Of The Review:**

This paper proposes a novel method, Average Thresholded Confidence (ATC) to learn a threshold on the model's confidence. It is well-written and well-motivated. The proposed idea is simple and technically sound. The claims are well-supported by theoretical analyses and extensive experimental results.

---

> ### Author Response · Authors · 2021-11-22
> **Response to Reviewer zA7T**
>
> Thanks for your positive feedback and for championing our paper. Please let us know if you have any questions that we could answer.
>
> We added some experiments and summarized them in the common response. If you do not have any questions, we hope that you might consider raising your score.

---

### Official Review · Reviewer_M2yw · 2021-11-02

**Correctness:** 4
**Technical Novelty And Significance:** 3
**Empirical Novelty And Significance:** 3
**Recommendation:** 8
**Confidence:** 4

**Main Review:**

-The proposed method is simple and easy to understand. I also think that their setting where no labeled data from the test distribution is available is highly relevant.

-Their evaluation contains a large number of datasets and distribution shifts and clearly demonstrates that ATC is superior to baseline methods. The only thing that is not quite clear is which scoring function should be used in practice, as there exist certain configurations where ATC-MC and ATC-NE perform the best.
They also compare to a regression model that is trained with labeled data and are able to achieve similar performance without this extra information.

-Is there an intuitive understanding why temperature rescaling helps? While temperature rescaling preserves the classification of the model, it can vary the ordering of the scores on different samples, however I would not have expected such large changes, especially in the order of ATC-NE pre and post T. Are there specific logit configurations that clearly show how temperature rescaling helps?

-I liked the initial theoretical discussion of the problem. The authors state that "every method of estimating accuracy on target data is tied up with some assumption on the nature of the shift", is it possible to elaborate on the assumptions of ATC in this context?



**Summary Of The Paper:**

The paper introduces Average Thresholded Confidence (ATC), a method that aims to predict a model's accuracy on an unlabeled test distribution that varies from the original training distribution by some form of distribution shift.
The paper first gives a theoretical analysis that shows that, in general, it is impossible to estimate test accuracy without assumptions on the particular distribution shift and that each method that tries to solve this problem has some underlying assumption on the distribution shift.

Their method ATC selects a threshold t on a scoring function (max confidence, or negative entropy) based on the error in the original source domain. The estimated error is then equal to the number of points in the test set that have a score lower than t. Importantly, this is done without using any label information from the test set.

In their experiments, they demonstrate improvements over various baseline methods on a large selection of datasets and distribution shifts.
Finally, they have a toy example with very strong assumptions on which ATC-MC is a consistent accuracy estimator.

**Summary Of The Review:**

The paper offers some good theoretical insight into the general problem and introduces a simple method that clearly outperforms the baselines. The problem statement is interesting and very relevant to the community. The evaluation is extensive and clearly demonstrates the strengths of this paper. While the theory does not cover all aspects (eg temperature rescaling), I liked the paper overall.

---

> ### Author Response · Authors · 2021-11-22
> **Response to Reviewer M2yw**
>
> We thank the reviewer for their positive and constructive feedback and for championing our paper.
>
> **”The only thing that is not quite clear is which scoring function should be used in practice, as there exist certain configurations where ATC-MC and ATC-NE perform the best.”**
>
> Across our experiments, we observed that in most of the datasets (all except ImageNet synthetic shifts) ATC-NE performs similarly or improves over ATC-MC. This can be attributed to using more information in the form of full softmax output instead of just the argmax confidence. More importantly, both the scoring functions improve over existing methods by a significant margin.
>
>
> **”Is there an intuitive understanding why temperature rescaling helps? While temperature rescaling preserves the classification of the model, it can vary the ordering of the scores on different samples.... Are there specific logit configurations that clearly show how temperature rescaling helps?”**
>
> We employ temperature scaling to calibrate classifier confidence to reflect its ground truth correctness likelihood as the score function uses the softmax output of the classifier. Since neural network predictions are known to be overconfident, temperature scaling or calibration, in general, reduces overestimation bias due to over-confident predictions. We hope to extend our theoretical model to multi-class classification and formalize this intuition to understand the efficacy of calibration.
>
> **”The authors state that "every method of estimating accuracy on target data is tied up with some assumption on the nature of the shift", is it possible to elaborate on the assumptions of ATC in this context?”**
>
> ATC is motivated by the superiority of methods that use maximum softmax probability (or logit) of a model for Out-Of-Distribution (OOD) detection (Hendrycks & Gimpel, 2016; Hendrycks et al., 2019). As described in Appendix C, our method can be interpreted as identifying examples for correct and incorrect prediction based on the value of the score function s(f(x)), i.e., if the score s(f(x)) is greater than or equal to the threshold t then our method predicts that the classifier correctly predicted datum (x,y) and vice-versa if the score is less than t.  A method that can solve this task will perfectly estimate the target performance.  However, such an expectation is unrealistic. Instead, with ATC, we require a relaxed condition. In particular, we want to obtain a threshold such that the number of falsely identified correct predictions match falsely identified incorrect predictions on source distribution, thereby balancing incorrect predictions. While most of the examples with a score above the threshold are correct and most of the examples below the threshold are incorrect, this condition allows accurate error prediction even with the separation is not perfect but we balance examples that are falsely identified as correct with the examples falsely identified as incorrect.
>
> As noted in the conclusions and future work section, we hope to investigate more general sufficient and necessary conditions in future work.

---

> > ### Comment · Reviewer_M2yw · 2021-11-29
> > **Response**
> >
> > I thank the authors for their helpful reply.
> >
> > While it is true that some further theoretical insights would have been helpful and the method is quite simple (which is not necessarily a bad thing), I think that the strong empirical results (also on the new experiments in the appendix) warrant acceptance.

---

### Official Review · Reviewer_uW7x · 2021-11-03

**Correctness:** 4
**Technical Novelty And Significance:** 3
**Empirical Novelty And Significance:** 3
**Recommendation:** 8
**Confidence:** 3

**Main Review:**

Overall, I find the paper quite interesting and intuitive. The mechanism is simple and seems to be effective compared to prior work. Given that this is an important real-world problem, I think the paper deserves to be accepted at ICLR.

However, I am not a 100% familiar with the related work in order to judge the novelty of the paper.

## Strengths

1. an appropriate amount of experiments
2. interesting discussion of related work
3. relevant theoretical results to motivate the problem

## Weaknesses and other Remarks

1. **Derivation and/or Interpretation of ATC**:
The authors did a decent job of providing some related theoretical results in the beginning. However, when the authors continue to introduce ATC this seems fairly ad-hoc without any kind of deeper explanation. I think this would merit a longer discussion.

2. **Proper contextualization of theoretical results**:
As I said before, I appreciate the theoretical results as motivating examples for considering empirical scores. However, I don't think they can be considered actual strong contributions of the paper. The main contributions are a heuristic-based score and accompanying experimental results showing the effectiveness. So please make sure that this is reflected in the writing.

3. **Clarification around some experimental settings**:
   a. Could you provide some confidence intervals around the scatter plot in Figure 2 and related figures, e.g., using bootstrapping. Why are there so many more sampled points around higher accuracies?
   b. Could you add more discussion around temperature scaling (cf. _For all methods, we implement post-hoc calibration on validation source data with Temperature Scaling_?) Why is this relevant and what's the intuition? If such an approach improves your performance, could also try improving your performance using more sophisticated calibration methods? That would be interesting to see.


## Ways to Improve My Score

Please address the weaknesses and I have listed above. I would be happy to engage in discussions during the rebuttal period.

---


## Update after rebuttal

The authors addressed my concerns in their responses. Consequently, I am raising my score from 6 to 8. My confidence score will remain at 3 given my missing knowledge about some of the prior work.

**Summary Of The Paper:**

The paper proposes a new scoring mechanism, Average Thresholded Confidence (ATC), to estimate out-of-distribution performance (i.e. accuracy) of a trained classifier using only labelled data from the source distribution (train distribution) and unlabelled data from the target distribution (test distribution).

The threshold is based on the logit output of a classifier and can thus be used on unlabelled data.


**Summary Of The Review:**

The paper is well-written and I enjoyed reading it. I cannot 100% verify the novelty of the paper since I am not so familiar with the related work but given what the authors wrote in their related work section and their results it seems novel.

Overall, I didn't have any major complaints about the paper.

---

> ### Author Response · Authors · 2021-11-22
> **Response to Reviewer uW7x**
>
> Thank you for your positive assessment and constructive feedback on our work. We are glad that you find our method a simple and effective solution to an important real-world problem.
>
> **Comment on novelty.** To the best of our knowledge, we are unaware of any prior work that proposes a similar method. We have compared our work with a bunch of very recently proposed methods (Chen et al., 2021; Guillory et al., 2021; Chuang et al., 2020, Deng & Zheng, 2021; Jiang et al., 2021; Deng et al., 2021). Additionally, recent work (Deng et al., 2021, Deng and Zheng, 2021) has also raised a question about a practical strategy to select a threshold that enables error prediction with thresholded model confidence and we believe our work takes a step in positively answering this question.
>
> **”Proper contextualization of theoretical results: As I said before, I appreciate the theoretical results as motivating examples for considering empirical scores... The main contributions are a heuristic-based score and accompanying experimental results showing the effectiveness.”**
>
> While we understand that our impossibility result is a minor contribution technically, it highlights the hardness of the problem that either (i) we need assumptions on the underlying classifier that renders accuracy estimation possible on target data, or (ii) we need assumptions on the nature of the shift. By its construction, ATC will consistently estimate accuracy on target distribution if and only if the number of incorrectly classified examples lying above the identified threshold matches the number of correctly classified examples that receive a score below the identified threshold (Appendix D.1). Overall, at a high level, our impossibility results motivate a classifier-dependent condition on the target data for consistent accuracy estimation.
>
> Moreover, we agree that the primary focus of the work is empirical. We make this clear in the updated version.
>
> **”Clarification around some experimental settings: a. Could you provide some confidence intervals around the scatter plot in Figure 2 and related figures, e.g., using bootstrapping? ”**
>
> Thanks for this suggestion. We have re-run experiments with 4 different seeds and updated all of our results. We observe no differences in relative superiority of ATC.
>
> **“b. Could you add more discussion around temperature scaling? Why is this relevant and what's the intuition? If such an approach improves your performance, could also try improving your performance using more sophisticated calibration methods? That would be interesting to see.”**
>
> We tune the single temperature parameter for TS calibration on labeled source data. Since the score function uses the softmax output of the classifier, we employ temperature scaling to calibrate classifier confidence to reflect its ground truth correctness likelihood.
> In our initial experiments, we tried other calibration methods like vector scaling. However, we didn’t observe any benefits of using vector scaling over temperature scaling. Hence, we primarily focused on temperature scaling in our experiments as temperature scaling doesn’t change the argmax prediction of the classifier.
>
> **”Derivation and/or Interpretation of ATC: The authors did a decent job of providing some related theoretical results in the beginning. However, when the authors continue to introduce ATC this seems fairly ad-hoc without any kind of deeper explanation....”**
>
> ATC is motivated by the superiority of methods that use maximum softmax probability (or logit) of a model for Out-Of-Distribution (OOD) detection (Hendrycks & Gimpel, 2016; Hendrycks et al., 2019). As described in Appendix C, our method can be interpreted as identifying examples for correct and incorrect prediction based on the value of the score function s(f(x)), i.e., if the score s(f(x)) is greater than or equal to the threshold t then our method predicts that the classifier correctly predicted datum (x,y) and vice-versa if the score is less than t.  A method that can solve this task will perfectly estimate the target performance.  However, such an expectation is unrealistic. Instead, with ATC, we require a relaxed condition. In particular, we want to obtain a threshold such that the number of falsely identified correct predictions match falsely identified incorrect predictions on source distribution, thereby balancing incorrect predictions. While most of the examples with a score above the threshold are correct and most of the examples below the threshold are incorrect, this condition allows accurate error prediction even with the separation is not perfect but we balance examples that are falsely identified as correct with the examples falsely identified as incorrect.
>
> As noted in the conclusions and future work section, we hope to investigate more general sufficient and necessary conditions in future work.

---

> > ### Author Response · Authors · 2021-11-23
> > **Response to Reviewer uW7x (cont.)**
> >
> > **”Why are there so many more sampled points around higher accuracies?”**
> >
> > We did not explicitly sample target datasets from higher accuracy regions but rather this is a characteristic property of distribution shifts in various datasets considered like CIFAR-10-C, ImageNet-10-C, etc. Intuitively, this implies that shifts in common corruptions (i.e., CIFAR-10-C and ImageNet-10-C) datasets show relatively smaller degradation in accuracy.

---

> ### Comment · Reviewer_uW7x · 2021-11-29
> **Concerns are addressed**
>
> Hi Authors,
> thank you for your response. My concerns are properly addressed at this point and I will raise my score accordingly (see update to my review above).

---

### Official Review · Reviewer_t6Wr · 2021-11-04

**Correctness:** 3
**Technical Novelty And Significance:** 2
**Empirical Novelty And Significance:** 3
**Recommendation:** 5
**Confidence:** 4

**Main Review:**

Strengths:
1. The experiments are thorough, conducted on a variety of types of shifts and models.

Weaknesses:
1. The proposed method lacks theoretical rigor. The threshold for source and target would differ based on different types of shifts, and there's no reason why the source threshold should be used to predict target accuracy. The fact that it works empirically is not a good justification.
2. No theoretical analysis are are done on NE vs. MC. When should we use one over the other, and why?
3. Section 3 doesn't fit with the rest of the paper. Since there's no general-purpose estimator, why does this paper still propose such an estimator? The impossible results are not surprising. It is perhaps more meaningful to study principled estimators under specific settings, just like the covariate shift or label shift settings.
4. I wish to see the theoretical analysis on the toy setting generalized. What are the necessary and sufficient conditions for ATC to work?
5. I wish the authors compare with other calibration methods beyond temperature scaling.

**Summary Of The Paper:**

This work proposes a simple method, Average Thresholded Confidence (ATC), to predict the OOD accuracy of a classifier based on a labeled source distribution and an unlabeled target distribution. First, the authors show a theoretical lower bound that such task is impossible without assumption on the nature of the shift. Then, the authors proposes ATC. Experiments show that ATC empirically performs better than baselines on different datasets and models. Theoretically, they show that ATC is a consistent estimator for a toy setting with spurious features.

**Summary Of The Review:**

The paper is strong empirically but lacks theoretical rigor.

---

> ### Author Response · Authors · 2021-11-22
> **Response to Reviewer t6Wr**
>
> We thank the reviewer for the thoughtful and detailed review.
>
> **“The proposed method lacks theoretical rigor. The threshold for source and target would differ based on different types of shifts, and there's no reason why the source threshold should be used to predict target accuracy. The fact that it works empirically is not a good justification.”**
>
> We do not argue that the source threshold can always be used to predict target accuracy. In fact, our impossibility result highlights that this method of estimating accuracy **not** work for all distribution shifts. Instead, in our empirical evaluation, we observe that the threshold obtained on source predicts target accuracy significantly better than previously proposed approaches on distribution shifts observed in practice across vision and natural language datasets.
>
> Our empirical finding together with the impossibility result hint that natural distribution shifts across different datasets satisfy the **same** simple structures that render ATC effective. This result is similar in spirit to the `accuracy on line' phenomena [1,2] observed in practice on these natural and synthetic shifts.
>
> **“No theoretical analysis is done on NE vs. MC. When should we use one over the other, and why?”**
>
> Across our experiments, we observed that in most of the datasets (all except ImageNet synthetic shifts) ATC-NE performs similarly or improves over ATC-MC. This can be attributed to using more information in the form of full softmax output instead of just the argmax confidence. In our current work, we consider the toy model for a binary classification problem where NE and MC have one-to-one mapping to one another and hence are the same. In the future, we hope to extend the analysis to multiclass problems.
>
> **“Section 3 doesn't fit with the rest of the paper. Since there's no general-purpose estimator, why does this paper still propose such an estimator? The impossible results are not surprising. It is perhaps more meaningful to study principled estimators under specific settings, just like the covariate shift or label shift settings.”**
>
> We want to make two clarifications about our work:
>
> (i) We do not claim ATC to be a general-purpose estimator.  As mentioned above, our main finding is that ATC estimates target accuracy significantly better than previously proposed approaches on distribution shifts observed in practice across vision and natural language datasets. In light of this empirical finding, our impossibility result highlights that natural distribution shifts across different datasets satisfy the **similar** simple structures that render ATC effective.
>
> Moreover, empirical comparisons with the importance re-weighting method from Mandolin (Chen et al. 2021) highlight that ATC might be more effective than methods tailored to tackle accuracy estimation under the covariate shift assumption.
>
> (ii) Our impossibility results establish the hardness of the problem where we highlight its equivalence with identifying the optimal target predictor. To the best of our knowledge, no prior work formalized these impossibility results. Additionally, we acknowledge that the primary contribution of our work is the method to estimate accuracy on target data and its empirical evaluation on a variety of distribution shifts.
>
>
> **”I wish to see the theoretical analysis on the toy setting generalized. What are the necessary and sufficient conditions for ATC to work?”**
>
> In our toy theoretical model, we show that ATC selects a threshold on source data such that it consistently predicts accuracy on both the sub-groups individually, i.e. the sub-group with spurious correlations and the sub-group without any spurious correlation. Hence, for all target shifts where we observe shifts in the relative population of these sub-groups, ATC consistently estimates target accuracy.
>
> In a similar vein, we expect ATC to work when (i) the source distribution can be decomposed as $\sum \alpha_i P_i $, where $\sum \alpha_i = 1$ and ATC estimates accuracy consistently on each $P_i$ and (ii) the distribution shift is considered such that we vary $\alpha_i$ by maintaining the constraint $\sum \alpha_i = 1$. Our toy model is a special case of this general framework where we theoretically show that ATC consistently estimates accuracy on each $P_i$.
>
> Conversely, our toy example also highlights a failure mode of ATC. Particularly when the population of each sub-group changes such that the threshold no longer estimates the accuracy consistently on individual sub-groups (Appendix D).

---

> > ### Author Response · Authors · 2021-11-22
> > **Response to Reviewer t6Wr (cont.)**
> >
> > **”I wish the authors compare with other calibration methods beyond temperature scaling.”**
> >
> > Indeed in our initial experiments, we tried other calibration methods like vector scaling. However, we didn’t observe any benefits of using vector scaling over temperature scaling. Hence, we primarily focused on temperature scaling in our experiments as temperature scaling doesn’t change the argmax prediction of the classifier.
> >
> > [1] Miller, John P., et al. Accuracy on the line: On the strong correlation between out-of-distribution and in-distribution generalization. International Conference on Machine Learning (ICML), 2021.
> >
> > [2] Recht, B., et al. Do ImageNet classifiers generalize to ImageNet? In International Conference on Machine Learning (ICML), 2019.

---

### Official Review · Reviewer_s1vP · 2021-11-05

**Correctness:** 4
**Technical Novelty And Significance:** 3
**Empirical Novelty And Significance:** 3
**Recommendation:** 8
**Confidence:** 4

**Main Review:**

Strengths
---
I am not an expert in this particular area, but to the best of my knowledge, the proposed technique is novel. Moreover, it is demonstrably more effective than prior methods and particularly simple to implement and use.

The paper is of relatively high quality. An extensive number of experiments are carried out on different distribution shift benchmarks, and it seems like the proper prior methods are compared to, though again, I am not an expert.

The paper is well-written and well-structured. This is aided in particular by the fact that the proposed technique is so simple to begin with.

Finally, the paper is likely to be of practical interest for researchers and practitioners studying and dealing with distribution shift.


Weaknesses
---
There are still a few substantive ways to improve the quality of the paper. First, it seems that neither of the theoretical sections (3 and 6) tie in well with the rest of the paper. In my view, the paper is primarily an empirical contribution. Section 3 ostensibly says nothing specific about the proposed technique, it just presents some general results that seem relatively obvious (though I am unsure whether they have been explicitly stated in prior work). But does the theory here motivate the proposed technique in any way? This is unclear to me. Section 6 is more closely related to the empirics, but this connection could still be improved. What are the key insights we can take away from the toy problem? How does it allow us to better understand for which real world problems the technique may succeed or fail? In my view, the paper could do a better job at addressing these questions.

Regarding empirics, the paper is significantly stronger here, but there is still some room for improvement. Most concretely, evaluations on the other WILDS datasets could be quite interesting. This is not (just) a generic "add more experiments" comment: all of the current experiments focus on natural (with the exception of some rendition and sketch shifts) object-centric images, and WILDS contains other modalities such as medical and satellite images, which makes for a rather interesting additional testbed. WILDS also contains language tasks, which would be separately interesting to evaluate on, though I acknowledge this may take more time to set up.

Regarding clarity, there are some minor ways to improve the paper. Section 4 could be expanded to provide more intuition and formal arguments for why the proposed technique makes sense and is justified, ideally from first principles. In fact, I may go as far as to say that this type of exposition would be more useful than Section 3 entirely, to the concerns I have noted above. Figures 2 and 3 were confusing to parse at first -- I think the lines are regression fits, but this should be made explicit if it currently isn't. Table 1 is a bit confusing in the first two columns because the datasets are listed but not necessarily the test sets. Finally, some of the comparisons such as IM should be explained in more detail.

**Summary Of The Paper:**

This paper proposes a simple yet effective technique for estimating a classifier's accuracy on test data which exhibits distribution shift from the training data. Given a score function for the model and a data point, the technique finds a threshold such that such that the relative proportions of validation data above and below the threshold are the same as correctly vs incorrectly classified points, respectively. This threshold is then used to predict the model's accuracy on test data, and the paper demonstrates that this predictor is significantly more accurate than prior methods on a range of distribution shift benchmarks. There are also some theoretical and empirical results on a toy problem that seem to indicate that the proposed technique is well-suited for at least this problem.

**Summary Of The Review:**

Primarily due to my concerns above, I am initially recommending a weak accept of this paper. I am happy to engage in discussion with the authors and other reviewers in order to reach a more confident final recommendation.

---

> ### Author Response · Authors · 2021-11-22
> **Response to Reviewer s1vP**
>
> We thank the reviewer for the constructive review and positive feedback. We are glad that you find our work of practical interest for researchers and practitioners studying and dealing with distribution shifts.
>
> **”Regarding empirics, the paper is significantly stronger here ... evaluations on the other WILDS datasets could be quite interesting. WILDS contains other modalities such as medical and satellite images, and contains language tasks.”**
>
> Thank you for your suggestion. Based on this, we have now added experiments on 4 WILDS datasets as mentioned in our common response. Note that while we included results on FMoW-WILDS in the first draft, our evaluation was restricted to distribution shifts observed due to temporal changes but aggregated across different countries. In the current version, we also evaluated different subgroups of target data as obtained by considering different countries individually.
>
> Similar to our results on other benchmarks, across all these WILDS datasets, ATC continues to outperform prior methods by a significant gap.
>
> **“It seems that neither of the theoretical sections (3 and 6) tie in well with the rest of the paper. In my view, the paper is primarily an empirical contribution … though I am unsure whether they (theoretical results from Section 3) have been explicitly stated in prior work.”**
>
> In Section 3, we establish impossibility results for the accuracy estimation where we highlight the hardness of the problem when we have no knowledge about the underlying classifier. To the best of our knowledge, no prior work formalized these impossibility results.
>
> Our results highlight that either (i) we need assumptions on the underlying classifier that renders accuracy estimation possible on target data or (ii) we need assumptions on the nature of the shift. By its construction, ATC will consistently estimate accuracy on target distribution if and only if the number of incorrectly classified examples lying above the identified threshold matches the number of correctly classified examples that receive a score below the identified threshold (Appendix D.1). At a high level, our impossibility results motivate a classifier-dependent condition on the target data for consistent accuracy estimation.
>
>
> **”Section 6 is more closely related to the empirics, but this connection could still be improved. What are the key insights we can take away from the toy problem? How does it allow us to better understand for which real world problems the technique may succeed or fail? In my view, the paper could do a better job at addressing these questions.”**
>
> In our toy theoretical model, we show that ATC selects a threshold on source data such that it consistently predicts accuracy on both the sub-groups individually, i.e. the sub-group with spurious correlations and the sub-group without any spurious correlation. Hence, for all target shifts where we observe shifts in the relative population of these sub-groups, ATC consistently estimates target accuracy.
>
> In a similar vein, we expect ATC to work when (i) the source distribution can be decomposed as $\sum \alpha_i P_i $, where $\sum \alpha_i = 1$ and ATC estimates error consistently on each $P_i$ and (ii) the distribution shift is considered such that we vary $\alpha_i$ by maintaining the constraint $\sum \alpha_i = 1$. Our toy model is a special case of this general framework where we theoretically show that ATC consistently estimates error consistently on each $P_i$.
>
> Conversely, our toy example also highlights a failure mode of ATC. Particularly when the population of each sub-group changes such that the threshold no longer estimates the accuracy consistently on individual sub-groups (Appendix D).

---

> > ### Author Response · Authors · 2021-11-22
> > **Response to Reviewer s1vP (cont.)**
> >
> > **“Section 4 could be expanded to provide more intuition and formal arguments for why the proposed technique makes sense and is justified, ideally from first principles. In fact, I may go as far as to say that this type of exposition would be more useful than Section 3 entirely, to the concerns I have noted above.”**
> >
> > ATC is motivated by the superiority of methods that use maximum softmax probability (or logit) of a model for Out-Of-Distribution (OOD) detection (Hendrycks & Gimpel, 2016; Hendrycks et al., 2019). As described in Appendix C, our method can be interpreted as identifying examples for correct and incorrect prediction based on the value of the score function s(f(x)), i.e., if the score s(f(x)) is greater than or equal to the threshold t then our method predicts that the classifier correctly predicted datum (x,y) and vice-versa if the score is less than t.  A method that can solve this task will perfectly estimate the target performance.  However, such an expectation is unrealistic. Instead, with ATC, we require a relaxed condition. In particular, we want to obtain a threshold such that the number of falsely identified correct predictions match falsely identified incorrect predictions on source distribution, thereby balancing incorrect predictions. While most of the examples with a score above the threshold are correct and most of the examples below the threshold are incorrect, this condition allows accurate error prediction even with the separation is not perfect but we balance examples that are falsely identified as correct with the examples falsely identified as incorrect.
> >
> > **“Figures 2 and 3 were confusing to parse at first -- I think the lines are regression fits, but this should be made explicit if it currently isn't.”**
> >
> > Yes, those lines are regression fits. We have made it clear in the updated draft.
> >
> > **”Table 1 is a bit confusing in the first two columns because the datasets are listed but not necessarily the test sets.”**
> >
> > We make the test datasets used clear in Table 2 on Page 21 in Appendix. We have now added a reference in Table 1.
> >
> > **“Finally, some of the comparisons such as IM should be explained in more detail.”**
> >
> > We include more details on IM baseline in Appendix E on page 20. We would be happy to include additional details if you have suggestions.

---

> > > ### Comment · Reviewer_s1vP · 2021-11-23
> > > **Thank you for your response**
> > >
> > > Thank you for your response, it resolves most of my concerns.
> > >
> > > The new results on more WILDS datasets are much appreciated. Why is there only one number for ATC for CivilComments?
> > >
> > > I also appreciate the additional exposition you have presented in your response for Section 4, but it looks like it hasn't found its way into the main paper. As you have acknowledged, the contribution of this work is primarily empirical. Additionally, reviewers t6Wr and uW7x have both raised similar concerns about the theoretical sections (particularly Section 3) and the brevity of Section 4. It is still my view that the paper would be improved and clarified by better explaining the empirical method in the main paper (e.g., using the exposition that is now in Appendix C) and finding space for this by moving (parts of) Section 3 to the appendix.

---

> > > > ### Author Response · Authors · 2021-11-24
> > > > **Thank you for your reply: Clarification on CivilComments and Theoretical sections**
> > > >
> > > > Thank you for your prompt reply. We greatly appreciate all the feedback so far, and we are glad our clarifications are helping!
> > > >
> > > > **Re. CivilComments**.  We have a single number for CivilComments because it is a binary classification task. For multiclass problems, ATC-NE and ATC-MC can lead to different ordering of examples when ranked with the corresponding scoring function. Temperature scaling on top can further alter the ordering of examples. The changed ordering of examples yields different thresholds and different accuracy estimates. However for binary classification, the two scoring functions are the same as entropy (i.e. $p\log(p) + (1-p) \log(p)$) has a one-to-one mapping to the max conf for $p\in [0,1]$. Moreover, temperature scaling also doesn't change the order of points for binary classification problems. Hence for the binary classification problems, both the scoring functions with and without temperature scaling yield the same estimates. We have made this clear in the updated draft.
> > > >
> > > > **Re. theoretical section.** Yes, we have added the text from Appendix C to Section 4 in the updated draft by adjusting text after Proposition 1 from Section 3.1 to Appendix.
> > > >
> > > > While we cannot update the draft currently, we will make sure that these changes are synced in the camera-ready version. Please let us know if there are any other questions that we can resolve.

---

> > > > > ### Comment · Reviewer_s1vP · 2021-11-24
> > > > > **Thanks!**
> > > > >
> > > > > Thanks for the additional clarifications and updates to the paper. I feel like my comments have been resolved and I am thus upgrading my recommendation.

---

### Author Response · Authors · 2021-11-22
**Common response**

We would like to thank the reviewers for their detailed and thoughtful feedback. We are glad to see that 4 reviewers recommend acceptance, noting the simplicity and practicality of the proposed method (s1vP, uW7x, M2yw, zA7T), our exhaustive empirical evaluation on different distribution shift benchmarks (s1vP, t6Wr, uW7x, M2yw, zA7T), and that the paper is well-written (s1vP, uW7x, M2yw, zA7T).

Inspired by the feedback from reviewers, we have run several experiments and added them to the draft. We summarize them below:

- **Additional experiments on WILDS benchmark**. We have now added experiments on 4 datasets from WILDS: FMoW (Satellite images), RxRx1 (cell image), Amazon (natural language sentiment), and CivilComments (natural language toxicity). Across all these datasets, we observe that *ATC continues to estimate target performance more accurately* as compared to prior methods.

- **More seeds for all the experiments**. As per the suggestion of uW7x, we have now included aggregate results from 4 different runs (i.e. with different seeds). We observe *no differences* in the relative superiority of ATC.

---

### Public Comment · ~Jiefeng_Chen1 · 2022-04-27
**A related work**

This paper is very related to our recent work, Detecting Errors and Estimating Accuracy on Unlabeled Data with Self-training Ensembles, published in NeurIPS 2021. Hope the authors could add some discussions about our work in the final camera-ready version of their paper.

---

### Decision · Program_Chairs · 2022-01-20

**Decision:**

Accept (Poster)

**Comment:**

The authors propose a simple method to estimate the accuracy of a classifier on an unlabeled dataset given an in-distribution validation set. In extensive experiments the authors show that the proposed method is significantly more accurate than previous methods and other baselines.

The reviewers are quite consistent in their judgement, just the weighting of the different aspects is different.
After the rebuttal four out of five reviewers recommend acceptance.

Strong points:
- simplicity of the method
- strong experimental results for various tasks and domain shift problems

Weak points:
- there is no clear theoretical statement when the method is supposed to work
- the discussion in Section 3.1 is pretty obvious and seems a bit like a waste of space whereas the motivation for the actual method is very short

While I agree with the reviewers that there is little theoretical justification for the method, the strong experimental results on various datasets, tasks and different domain shifts make this paper interesting for a large audience. Thus this paper is a nice contribution to ICLR and I recommend acceptance.
However, I strongly recommend to the authors to add more motivation in Section 4 and add a limitation section where the cases are discussed where the method is definitely not working. Section 3 is pretty obvious and could be significantly shortened or integrated into the limitations section.

One case which is highly relevant for this limitations section is the provable asymptotic overconfidence of neural networks which is discussed in

Hein et al, Why ReLU networks yield high-confidence predictions far away from the training data and how to mitigate the problem, CVPR 2019

This would definitely lead to a failure of the presented method as all predictions would get a score above the threshold. I would also assume that the method would predict high accuracy values for out-of-distribution tasks which are semantically similar e.g. training on CIFAR10 and then using CIFAR100 as unlabeled dataset. In that context it would be interesting to evaluate OOD-aware classifiers using ATC such as discussed in

Hendrycks et al, Deep Anomaly Detection with Outlier Exposure, ICLR 2019

Also it would be helpful to understand better the influence of the classifier performance on the original task on the performance of ATC on unlabeled data.